



# Shipborne measurements of ClNO₂ in the Mediterranean Sea and around the Arabian Peninsula during summer

Philipp G. Eger[1], Nils Friedrich[1], Jan Schuladen[1], Justin Shenolikar[1], Horst Fischer[1], Ivan Tadic[1], Hartwig Harder[1], Monica Martinez[1], Roland Rohloff[1], Sebastian Tauer[1], Friederike Fachinger[2], Frank Drewnick[2], James Brooks[3], Eoghan Darbyshire[3], Jean Sciare[4], Michael Pikridas[4], Jos Lelieveld[1], and John N. Crowley[1]

[1]Atmospheric Chemistry Department, Max-Planck-Institute for Chemistry, 55128 Mainz, Germany
[2]Particle Chemistry Department, Max-Planck-Institute for Chemistry, 55128 Mainz, Germany
[3]Centre for Atmospheric Science, University of Manchester, UK
[4]Energy, Environment and Water Research Center, The Cyprus Institute, Nicosia 1645, Cyprus

*Correspondence to*: John N. Crowley (john.crowley@mpic.de)

**Abstract.** Shipborne measurements of nitryl chloride (ClNO₂), hydrogen chloride (HCl) and sulphur dioxide (SO₂) were made during the AQABA (Air Quality and climate change in the Arabian BAsin) ship campaign in summer 2017. The dataset includes measurements over the Mediterranean Sea, the Suez Canal, the Red Sea, the Gulf of Aden, the Arabian Sea, the Gulf of Oman and the Arabian Gulf (also known as Persian Gulf) with observed ClNO₂ mixing ratios ranging from the limit of detection to ≈ 600 pptv. We examined the regional variability in the generation of ClNO₂ via the uptake of dinitrogen pentoxide (N₂O₅) to Cl-containing aerosol and its importance for Cl-atom generation in a marine boundary layer under the (variable) influence of emissions from shipping and oil industry. The yield of ClNO₂ formation per NO₃ radical generated was generally low (median of ≈ 1–5 % depending on the region), mainly as a result of gas-phase loss of NO₃ dominating over heterogeneous loss of N₂O₅, the latter being disfavoured by the high temperatures found throughout the campaign. The contributions of ClNO₂ photolysis and OH-induced HCl oxidation to Cl-radical formation were derived and their relative contributions over the diel cycle compared. The results indicate that over the northern Red Sea, the Gulf of Suez and the Gulf of Oman the formation of Cl-atoms will enhance the oxidation rates of some VOCs, especially in the early morning.

## 1 Introduction

The AQABA (Air Quality and climate change in the Arabian BAsin) campaign was designed to study air quality and climate in a region (Eastern Mediterranean and Middle East) that is likely to be heavily impacted by future climate change with increasing frequency and intensity of droughts, heatwaves and associated Aeolian dust and pollution emissions (Lelieveld et al., 2012). As the Arabian Gulf already suffers from some of the most polluted air on earth with O₃ levels regularly greater than 100 ppbv (Lelieveld et al., 2009), one aspect of the campaign was to investigate the factors that contribute to high levels





of air pollution in the region. This includes the impact of reactive chlorine chemistry resulting from the interactions of pollutant emissions from ships and petrochemical activity with sea-salt, under conditions influenced by intense photochemistry and high temperatures during summer.

The heterogeneous uptake of gaseous $N_2O_5$ to the aerosol phase represents an important atmospheric sink for $NO_x$ ($NO_x =$ NO + $NO_2$) via conversion to nitric acid ($HNO_3$), which is efficiently removed from the boundary layer via deposition (Lelieveld and Crutzen, 1990; Dentener and Crutzen, 1993; Macintyre and Evans, 2010). In the presence of aerosol chloride, nitryl chloride ($ClNO_2$) can also be formed along with $HNO_3$ ($NO_3^-$) as shown in Reaction (R1) (Finlayson-Pitts et al., 1989; Behnke et al., 1997). $ClNO_2$ has a lifetime of more than 30 hours in the nocturnal marine boundary layer (Osthoff et al., 2008) but is rapidly photolysed after sunrise (Reaction R2), releasing nitrogen dioxide ($NO_2$) and chlorine atoms.

$N_2O_5$ + ($H_2O$ or $Cl^-$)        $\rightarrow$        (2-$f$) $NO_3^-$ + $f$ $ClNO_2$,        $0 \leq f \leq 1$        (R1)

$ClNO_2$ + $h\nu$        $\rightarrow$        Cl + $NO_2$        (R2)

The formation of $ClNO_2$ can have a significant impact on regional $NO_x$ cycling and radical chemistry especially in the polluted coastal and marine boundary layer (Simon et al., 2009; Riedel et al., 2014; Sarwar et al., 2014). The Cl-atoms formed in Reaction (R2) can enhance oxidation rates of several volatile organic compounds (VOCs) especially during early

morning hours (Phillips et al., 2012; Riedel et al., 2012a; Young et al., 2012) thus contributing to photochemical ozone production (Simon et al., 2009; Riedel et al., 2014; Sarwar et al., 2014; Faxon et al., 2015; Wang et al., 2019).

The chemical processes involved in the formation of $ClNO_2$ are complex and, as outlined in Fig. 1, involve the sequential oxidation of $NO_x$ to $N_2O_5$ via $NO_3$ (Reactions R3–R5). During the day $NO_3$ is rapidly photolysed via Reaction (R6) or reacts with nitrogen oxide (NO) via Reaction (R7) so that $N_2O_5$ formation is supressed. The heterogeneous reaction of $N_2O_5$ with

particles is thus, to a good approximation, limited to the night-time. The equilibrium between $NO_3$ and $N_2O_5$ (Reactions R4 and R5) is strongly temperature-dependent, with $N_2O_5$ formation favoured by high $NO_2$ mixing ratios and low temperatures. $NO_3$ can also react with volatile organic compounds (VOCs) (R8) forming e.g. alkyl nitrates, which also reduces the rate of formation of $N_2O_5$.

$NO_2$ + $O_3$        $\rightarrow$        $NO_3$ + $O_2$        (R3)

$NO_3$ + $NO_2$        $\rightarrow$        $N_2O_5$        (R4)

$N_2O_5$ + M        $\rightarrow$        $NO_3$ + $NO_2$ + M        (R5)

$NO_3$ + $h\nu$        $\rightarrow$        NO + $O_2$        (R6)

$NO_3$ + NO        $\rightarrow$        2 $NO_2$        (R7)

$NO_3$ + VOCs        $\rightarrow$        products        (R8)

The $N_2O_5$ loss rate via heterogeneous uptake to particles is described by Eq. (1) where $\bar{c}$ is the average molecular velocity of $N_2O_5$, $A$ is the particle surface area concentration and $\gamma$ is the uptake coefficient.

$$\frac{d[N_2O_5]}{dt} = -0.25\,\bar{c}\,\gamma\,A\,[N_2O_5]$$        (1)



The uptake coefficient, $\gamma$, has been characterised in several laboratory investigations (see Bertram and Thornton (2009); Chang et al. (2011); Ammann et al. (2013) for summaries) and in numerous field studies where it has been found to be highly variable (between $5 \times 10^{-4}$ and 0.11) and dependent on temperature, relative humidity (RH) and aerosol composition (Brown et al., 2006; Bertram et al., 2009; Brown et al., 2009; Riedel et al., 2012b; Wagner et al., 2013; Morgan et al., 2015;

Brown et al., 2016; Phillips et al., 2016). A value of $\approx 0.03$ has been derived from measurements in the polluted marine environment (Aldener et al., 2006).

The $ClNO_2$ yield, $f$, which controls the relative formation rates of $NO_3^-$ and $ClNO_2$ in Reaction (R1), is determined by the $[Cl^-]$ to $[H_2O]$ ratio in the aerosol phase (Behnke et al., 1997; Bertram and Thornton, 2009; Ammann et al., 2013), and can vary between zero to unity (Thornton et al., 2010; Wagner et al., 2012; Riedel et al., 2013; Phillips et al., 2016; Wang et al.,

2016; McDuffie et al., 2018b). In Fig. 1 we introduce the $ClNO_2$ production efficiency $\varepsilon$, which is the yield of $ClNO_2$ per $NO_3$ molecule formed in Reaction (R3) and will be discussed in detail in Sect. 3.2.

The established method to measure atmospheric $ClNO_2$ mixing ratios from a few tens of pptv (part per trillion by volume) to several ppbv (parts per billion by volume) is Chemical Ionisation Mass Spectrometry (CIMS) using iodide ions to generate $IClNO_2^-$ which can be detected at a mass-to-charge ratio ($m/z$) of 208 and 210 (McNeill et al., 2006). The first measurement

highlighting the importance of $ClNO_2$ in the polluted marine boundary layer was performed by Osthoff et al. (2008) who detected mixing ratios exceeding 1 ppbv along the coast of Houston, Texas, originating from ship-plumes and urban and industrial $NO_x$ sources. This was the starting point for numerous measurements of $ClNO_2$ in various locations around the globe with an initial focus on coastal areas in the United States (U.S.), e.g. the Los Angeles Basin in California (Riedel et al., 2012a; Wagner et al., 2012; Young et al., 2012). Other studies included coastal sites in Canada (Osthoff et al., 2018) and

coastal / urban sites in the United Kingdom (Bannan et al., 2015; Bannan et al., 2017; Priestley et al., 2018; Sommariva et al., 2018). Whereas $ClNO_2$ was initially believed to play a significant role only in areas with marine influence (Behnke et al., 1997; Keene et al., 1999), mid-continental measurements in the U.S. (Thornton et al., 2010; Riedel et al., 2013; Faxon et al., 2015) revealed the importance of anthropogenic sources (e.g. industrial combustion, cooling towers, natural gas extraction and suspension of road salt) and sea salt chloride transported inland. Further studies reported significant mixing ratios of

$ClNO_2$ at a semi-rural site in continental Germany (Phillips et al., 2012; Phillips et al., 2016) and at a mid-continental urban site in Canada (Mielke et al., 2011; Mielke et al., 2016). Observations at continental sites could be reproduced by a global model (Wang et al., 2019) when considering the transport of HCl (aq) which had been initially formed in the gas-phase through acid displacement in coastal regions. More recently, $ClNO_2$ at the $> 1$ ppbv level has been observed in the heavily industrialised North China Plain (Tham et al., 2016; Liu et al., 2017; Wang et al., 2017; Tham et al., 2018), with even larger

mixing ratios measured in Beijing (Le Breton et al., 2018; Zhou et al., 2018) and Hong Kong (Wang et al., 2016).

The great variability seen in $ClNO_2$ mixing ratios in different locations reflects regional variability in its efficiency of production, which, as described above involves a complex set of chemical reactions, both in the gas- and particle phase and which will vary over time and space. Most measurements of $ClNO_2$ to date have been measurements at single locations, though some data from mobile platforms such as aircraft (Mielke et al., 2013; Lee et al., 2018; McDuffie et al., 2018a;





McDuffie et al., 2018b) and ships (Kercher et al., 2009; Riedel et al., 2012a) are available. With respect to understanding the formation and role of ClNO$_2$, much of the atmospheric boundary layer remains unexplored.

Here we present shipborne measurements of ClNO$_2$ in the marine boundary layer of the Mediterranean Sea and around the Arabian Peninsula, including the Red Sea and the Arabian Gulf. With a ship track from southern France to Kuwait we

provide a unique marine ClNO$_2$ dataset with a large spatial coverage. This allows us to investigate the ClNO$_2$ production efficiency ε and its regional impact under various atmospheric conditions ranging from polluted marine and coastal environment to low-NO$_x$ conditions in chemically aged air masses.

## 2 Methods

### 2.1 AQABA campaign

The measurements presented in this study were performed during the AQABA campaign which took place along the sea route between southern France and Kuwait in summer 2017. Five air-conditioned measurement containers with a variety of gas-phase and aerosol instrumentation were set up on-board the research vessel *Kommander Iona* which departed from Southern France on the 24$^{th}$ of June and passed various regions including the Mediterranean Sea, the Suez Canal, the Red Sea, the Gulf of Aden, the Arabian Sea, the Gulf of Oman and the Arabian Gulf (see Fig. 2), reaching its destination Kuwait

on the 31$^{st}$ of July (first leg) and covering a latitude / longitude span of 12–43 °N and 6–60 °E. After a short break in Kuwait the ship returned via the same route to southern France, arriving on the 2$^{nd}$ of September (second leg). The trace-gases described in this paper were sampled from the centre of a common, high volume-flow inlet (10 m$^3$ min$^{-1}$, 0.15 m in diameter, 0.2 s residence time) made of stainless steel, which was located on a measurement container at the front of the ship at a height of approximately 5.5 meters above the foredeck.

Depending on the wind direction relative to the movement of the vessel, measurements were occasionally impacted by emissions from the stack of our own ship. Especially on the first leg, the relative wind direction was frequently from behind where the chimney was located. All datasets were filtered prior to analysis for periods where the measurements were contaminated by stack emissions. The filter is based on short-term variation in NO and SO$_2$ signals and relative wind direction and reduces the useful data coverage to 58 % on the first leg and 95 % on the second leg.

### 2.2 Measurement of ClNO$_2$, HCl and SO$_2$

Nitryl chloride (ClNO$_2$), hydrogen chloride (HCl) and sulphur dioxide (SO$_2$) were detected with a Chemical Ionisation Quadrupole Mass Spectrometer (CI-QMS) using an electrical, radio-frequency (RF) discharge ion-source. The instrument and the ion-molecule-reactions involved in the detection of the above-mentioned trace gases are described in detail by Eger et al. (2019). Briefly, ClNO$_2$ was monitored as IClNO$_2^-$ at a mass-to-charge ratio (*m/z*) of 208 and 210 subsequent to the

reaction of ClNO$_2$ with I$^-$ (McNeill et al., 2006; Osthoff et al., 2008; Thornton et al., 2010). IClNO$_2^-$ is more specific than ICl$^-$ (*m/z* 162 and 164) and has a lower background signal, providing a sensitivity of 0.61 Hz pptv$^{-1}$ per 10$^6$ Hz of I$^-$, a limit





of detection (LOD) ($2\sigma$, 5 min) of 12 pptv and a total measurement uncertainty of 30 % $\pm$ 6 pptv. HCl was observed as I(CN)Cl$^-$ (*m/z* 188 and 190) with a sensitivity of 0.17 Hz pptv$^{-1}$ per $10^6$ Hz of I$^-$, a detection limit of 98 pptv and a total measurement uncertainty of 20 % $\pm$ 72 pptv. SO$_2$ was detected as ISO$_3^-$ (*m/z* 207) with a sensitivity of 0.10 Hz pptv$^{-1}$ per $10^6$ Hz of I$^-$, a detection limit of 38 pptv and a total uncertainty of 20 % $\pm$ 23 pptv.

A flow of 2.5 slm (standard litres per minute) was drawn into the CI-QMS instrument via a $\approx$ 3 m long ¼ inch PFA tubing while a 20 cm section of the inlet line in front of the IMR (ion molecule reactor) was heated to 200 °C to enable detection of peroxyacetyl nitrate (PAN) which is not reported here. An additional bypass (1 slm) in front of the IMR pinhole was installed to improve response times. The IMR region was held at a pressure of 18.00 $\pm$ 0.05 mbar by a dry vacuum scroll pump. The background signal was determined by periodically bypassing ambient air through a scrubber filled with steel

wool where the trace gases of interest are efficiently destroyed at the hot surfaces (120 °C). To avoid condensation of water in the inlet lines in the containers, the pressure in the sampling line was reduced by including an additional $\approx$ 50 cm long piece of 1/8 inch PFA tubing. A 2 µm pore size membrane filter (Pall Teflo) was placed between high volume-flow inlet and CI-QMS sampling line to remove particles and was exchanged regularly to avoid accumulation of particulate matter. No indication for ClNO$_2$ formation via N$_2$O$_5$ reactions on salty surfaces in the inlet line was observed during AQABA, i.e.

whenever we changed the particle filter or the inlet line, no change in signal was observed. Further the ClNO$_2$-to-N$_2$O$_5$ ratio was highly variable during AQABA (range of 0.35–59 with a median of 3.2) and ClNO$_2$ was occasionally measured in periods where no N$_2$O$_5$ was present.

ClNO$_2$ was calibrated twice during the campaign by simultaneously sampling a source of ClNO$_2$ via the CI-QMS and by a thermal dissociation cavity ring-down spectrometer (Sobanski et al., 2016). ClNO$_2$ was generated by passing Cl$_2$ over

NaNO$_2$ as described previously (Thaler et al., 2011; Eger et al., 2019). The signals at *m/z* 208 and *m/z* 210 showed a correlation of $R^2 = 0.93$ during ambient measurements ($R^2 = 0.99$ during calibrations).

HCl was calibrated four times throughout the campaign by adding a small flow over a permeation source to the main flow and monitoring the CI-QMS signal at *m/z* 188 and 190. SO$_2$ calibrations were performed seven times during the AQABA campaign by addition of a known flow of SO$_2$ from a gas cylinder (1 ppmv in synthetic air, Air Liquide). In contrast to

ClNO$_2$ and HCl, correction of the SO$_2$ signal for its relative humidity (RH) dependence was necessary, which we derived from calibrations during AQABA where the RH was actively varied between 1 and 80 %.

The CI-QMS was operated in selected ion monitoring mode measuring mainly ClNO$_2$, HCl, SO$_2$, PAN and peracetic / acetic acid with a temporal resolution of approximately 15 s for each molecule. Changes in sensitivity were captured by permanently monitoring the primary ion signal (I$^-$ and its water cluster) during ambient measurements and a background

signal was recorded every 100 minutes. For further analysis, all data sets were averaged to 5 min temporal resolution. Our ClNO$_2$, HCl and SO$_2$ datasets provide about 12,500 data points distributed over 61.4 days with interruptions due to background determinations, calibrations, filter and gas bottle changes and instrument power-down at the harbours of Jeddah and Kuwait. For periods where the ship was in motion, the data coverage for all three trace gases was about 80 %.



### 2.3 Other trace gases

O$_3$ was measured by a commercial ozone monitor (2B Technologies, Model 202) based on optical absorption at 254 nm with a detection limit of 3 ppbv (10 s) and a total uncertainty of 2 % ± 1 ppbv. Mixing ratios of NO$_x$ and NO$_y$ (NO$_y$ = NO$_x$ + reactive nitrogen trace gases + particulate nitrate) were monitored via Thermal Dissociation Cavity Ring-Down

Spectroscopy (TD-CRDS) using a modified version of the instrument described by Thieser et al. (2016). The difference between the NO$_y$ and the NO$_x$ signal is referred to as NO$_z$, which includes organic nitrates (peroxyacetyl nitrates and alkyl nitrates), NO$_3$, N$_2$O$_5$, ClNO$_2$, HNO$_3$ and particulate nitrate. In contrast to Thieser et al. (2016), the TD-unit was operated at 850 °C to ensure detection of HNO$_3$ and nitrate in the particle phase. The detection limits for NO$_x$ and NO$_y$ were 80 and 160 pptv, respectively, with total uncertainties of 9 % ± 30 pptv. NO$_z$ was calculated from measured NO$_x$ and NO$_y$ with a

detection limit of 160 pptv and a total uncertainty of 13 % ± 42 pptv. NO$_2$ (LOD = 52 pptv (1s), total uncertainty = 7%) and N$_2$O$_5$ (LOD = 6 pptv (1s), total uncertainty = 15%) were measured by a five-channel TD-CRDS described by Sobanski et al. (2016). NO and NO$_2$ were measured by a modified commercial Chemiluminescence Detector (CLD 790 SR) (ECO Physics, Duernten, Switzerland) (Fontijn et al., 1970). The LOD (5 s) was 21 pptv for NO and 52 pptv for NO$_2$ and the total uncertainty 6 % respectively 7 %. The NO$_2$ data was in good agreement with the CRDS dataset (R² = 0.95) with a mean

deviation of 6 %. The hydroxyl radical (OH) was measured using a Laser-Induced-Fluorescence method (Martinez et al., 2010; Novelli et al., 2014).

### 2.4 Meteorological parameter and actinic flux

Photolysis rates ($J_{O(1D)}$, $J_{ClNO2}$ and $J_{NO3}$) were calculated from wavelength resolved actinic flux measured by a spectral radiometer (Metcon GmbH; Meusel et al. (2016)) located close to the common trace-gas inlet. Cross sections and quantum

yields were taken from Burkholder et al. (2015). A commercial NEPTUNE weather-station (Sterela) monitored various parameters such as temperature, relative humidity, wind speed and direction, speed of the vessel and GPS position.

### 2.5 Aerosols

An Aerosol Mass Spectrometer (Aerodyne HR-ToF-AMS, DeCarlo et al. (2006)) measured PM$_1$ non-refractory aerosol composition (30 s time resolution), including sulphate, nitrate, ammonium, chloride and total organics with an overall

uncertainty of 35 %. An Optical Particle Spectrometer (OPC, Grimm model 1.109) measured the size distribution from 250 nm to 32 µm (6 s time resolution) with a total uncertainty of 25 %. A Fast Mobility Particle Spectrometer (FMPS, TSI model 3091) provided particle size distributions from 5.6 nm up to 560 nm (1 s time resolution). The particle surface area concentrations for PM$_1$ and PM$_{10}$ were calculated from the OPC and FMPS datasets, the overall uncertainty of these variables is estimated to be 30 %. The inlet for the aerosol instrumentation was located at the top of a measurement container

at a distance of ≈ 5 m to the common, trace-gas inlet described above. In order to avoid condensation in inlet lines, aerosol samples were passed through a drying system which reduced ambient relative humidity to an average value of ≈ 40 % in the





measurement container. We calculated the ambient $PM_1$ particle surface area concentration ($A$) from the measured surface area concentration using a hygroscopic growth factor (on average $1.32 \pm 0.24$) based on ambient RH and aerosol composition. The calculation of the growth-factor is described in the supplementary information (Figures S1–S4).

The water soluble fraction of total suspended particles (TSP) was monitored with hourly resolution using a Monitor for AeRosols and Gases in Ambient Air, MARGA (Metrohm Applikon Model S2), sampling at a distance of $\approx 5$ m from the common gas-phase inlet. In this work only results from $Na^+$ and $Cl^-$ measurements (TSP), with detection limits equal to 0.05 and 0.01 µg m$^{-3}$, will be used.

## 3 Results and discussion

In the following, we use only data (5 min averages) which were free from contamination by the ship's own exhaust.

### 3.1 Overview of measurement regions and $ClNO_2$ mixing ratios observed

Figure 2 illustrates the ship's track during AQABA, divided into seven regions demarked by dashed black lines: The Mediterranean Sea, the Suez Canal including the Gulf of Suez, the Red Sea, the Gulf of Aden, the Arabian Sea, the Gulf of Oman and the Arabian Gulf. On the first leg, the CI-QMS measurements started south of Crete; on the second leg measurements terminated close to Sicily after two months of almost continuous measurement. The 5 min averaged, maximum $ClNO_2$ mixing ratios observed during each night ranged from the limit of detection to 586 pptv. Figure 2 (text boxes) also indicates the median night-time mixing ratios of $O_3$, HCl, $NO_2$ and $SO_2$ for the different regions where data from the first and second leg datasets have been combined. The night-time mean, median and range of the mixing ratios of these trace gases (and also of $NO_2$, temperature, relative humidity, $NO_3$ production rate and $PM_1$ particle surface area concentration) are listed in Table 1; a time-series of measured $ClNO_2$, HCl, $SO_2$ and $O_3$ is provided in Fig. S5 of the supplementary information. The predominant air-mass origin for each night was derived from 48 h back-trajectories calculated with HYSPLIT (Stein et al., 2015; Rolph et al., 2017) and is illustrated for both legs in Fig. S6 of the supplementary information. While Fig. 2 provides an overview of the measurements during both legs, Fig. 3a and 3b highlight 9-day periods indicating features that characterised the transition from one region to the next. Based on these three figures, we will discuss observed $ClNO_2$ mixing ratios and related parameters for the seven regions defined above.

Over the Mediterranean Sea, during periods when the CIMS was operational, we encountered mainly aged air masses which had passed over Italy, Greece or Turkey (Fig. S6) characterised by relatively high $O_3$ levels but low $NO_2$-to-$NO_y$ ratios. As illustrated in Fig. 1 the formation of $ClNO_2$ is initiated by $NO_3$ production which will depend on $O_3$ levels and availability of $NO_2$. For the Mediterranean Sea the low $NO_2$ mixing ratios resulted in a weak $NO_3$ production term (Table 1) and low $ClNO_2$ mixing ratios. The only exceptions are two nights south of Sicily on the second leg where $ClNO_2$ mixing ratios up to 439 pptv were observed, which coincided with an increase in $NO_2$ originating from industrial sources on the mainland (Sicily and Italy). There are no previous measurements of $ClNO_2$ over the Mediterranean Sea but our data can be compared to the





output of a regional model (Li et al., 2019) which predicts monthly average $ClNO_2$ mixing ratios up to 100 pptv in the south-eastern Mediterranean Sea and around Sicily, in  broad agreement with our observations.

The Suez Canal and the Gulf of Suez were impacted by fresh emissions from ships, industry and urban centres with high $NO_2$, $SO_2$ and HCl mixing ratios. On the night from the $22^{nd}$ to the $23^{rd}$ of August we measured the highest $ClNO_2$ mixing
ratio of the whole campaign (586 pptv) due to exceptionally high $NO_2$ levels and $NO_3$ production rates.

Over the Red Sea, $O_3$ levels were elevated with the highest $NO_3$ production rates (up to 0.7 pptv $s^{-1}$) observed when approaching the Gulf of Suez with $NO_x$ transported south from the region around the Suez Canal, the city of Cairo and the Sinai Peninsula (see back-trajectories in Fig. S6). $ClNO_2$ mixing ratios exceeded 200 pptv every night (with a maximum of 480 pptv) whereby elevated $PM_1$ particle surface area concentration and HCl mixing ratios indicated increased
heterogeneous uptake and chloride availability.

Over the Gulf of Aden, with air mainly originating from Somalia, $O_3$ levels were close to 25 ppbv and $NO_2$ mixing ratios regularly exceeding 5 ppbv resulted in a $NO_3$ production rate up to ≈ 0.2 pptv $s^{-1}$. However, the $PM_1$ particle surface area concentration remained low and $ClNO_2$ was detected only occasionally (maximum of 379 pptv).

Over the Arabian Sea we experienced strong winds from the south with 48 h back-trajectories touching the coast of Somalia.
$ClNO_2$ was generally below the detection limit (maximum 56 pptv) and $NO_2$, HCl and $PM_1$ particle surface area concentration were very low. Missing local sources of $NO_x$ and low $O_3$ mixing ratios resulted in a weak $NO_3$ production term, partially responsible for the lack of $ClNO_2$. Low mixing ratios of $ClNO_2$ were occasionally detected that originated from single ships or point sources on the mainland.

Upon entering the Gulf of Oman, which marks the transition between remote marine environment and increased emissions
from petrochemical industry and shipping lanes, $NO_3$ production rates increased significantly due to higher $NO_x$ and $O_3$ levels. $ClNO_2$ mixing ratios exceeded 200 pptv during two consecutive nights with a maximum value in this region of 376 pptv.

The Arabian Gulf was characterised by very high ozone levels (sometimes exceeding 150 ppbv) and $SO_2$ mixing ratios that generally exceeded 5 ppbv. For the first leg (sailing into the Arabian Gulf), the air-mass passed over Kuwait whereas for the
second leg (sailing out of the Arabian Gulf) it mainly passed over Iran. In the Gulf region, which was heavily polluted by emissions from shipping and petrochemical industry, we also observed the highest HCl and $PM_1$ particle surface area concentration throughout the whole campaign. However, despite high $NO_3$ production rates (≈ 0.4 pptv $s^{-1}$) due to $NO_x$ emissions from oil and gas refineries as well as emissions from shipping and urban areas, we only observed relatively low $ClNO_2$ mixing ratios with a maximum value of 126 pptv close to Kuwait.
Consistent with Osthoff et al. (2008) we find significant amounts of nocturnal $ClNO_2$ in aged ship-plumes that could be identified by a defined peak-shape (Fig. S7) and covariance of $NO_2$ and $SO_2$ indicative of upwind point sources. As $SO_2$ and $NO_2$ are co-emitted from the combustion of ship fuel, it is not surprising that they show a co-variance. The consequence of the co-emission of $NO_2$ and $SO_2$ is that $ClNO_2$ is generally observed in the presence of both whereby high $ClNO_2$ mixing ratios were associated with aged ship-plumes.



Figure 4 shows diel profiles of nitryl chloride for the Red Sea and the Gulf of Oman together with the photolysis rate constant $J_{ClNO_2}$ illustrating that mixing ratios generally decreased at sunrise with a $ClNO_2$ lifetime of a few hours. Diel $ClNO_2$ profiles for other regions (see Fig. S8 of the supplementary information) look generally similar but with a varying maximum mixing ratio. Over the Red Sea $ClNO_2$ was often observed in plumes, whereas mixing ratios over the Gulf of

Oman increased continuously after sunset indicating that we sampled a more homogeneously polluted air mass in which $ClNO_2$ accumulated over the course of the night. The median mixing ratio in the afternoon was still around 10 pptv, which we attribute to an HCl interference at $m/z$ 208 and 210 described by Eger et al. (2019) rather than to the presence of $ClNO_2$ during the day when its production rate is close to zero and its lifetime is short due to photolysis. The magnitude of the HCl interference at the $m/z$ used to monitor $ClNO_2$ was derived during HCl calibrations on-board the ship and found to be 0.006

Hz (pptv of HCl)$^{-1}$ which is about 1 % of the $ClNO_2$ count rate of 0.61 Hz (pptv of $ClNO_2$)$^{-1}$ at $10^6$ Hz of I$^-$. However, during ambient air measurements the interfering signal was variable with a campaign average of $0.008 \pm 0.005$ Hz pptv$^{-1}$, which implies that a correction based on the HCl signal alone is not sufficient. The variable offset at the $ClNO_2$ mass contributes to the total measurement uncertainty and can be significant when analysing data close to the detection limit.

Although on occasions several hundred pptv of $ClNO_2$ were observed, below we show that the $ClNO_2$ production efficiency

was generally low. Reasons for this are examined in the following sections.

### 3.2 ClNO₂ yield per NO₃ molecule formed

We define the $ClNO_2$ production efficiency ($\varepsilon$) during AQABA as the number of $ClNO_2$ molecules generated per $NO_3$ molecule formed from the reaction of $NO_2$ with $O_3$ (Reaction R3). The instantaneous production rate of $NO_3$ is given by $k_1[NO_2][O_3]$ and the total number of $NO_3$ molecules formed over the course of the night is derived using a rate coefficient of

$k_1 = 1.4 \times 10^{-13}$ exp (-2470/T) cm$^3$ molecule$^{-1}$ s$^{-1}$ (IUPAC, 2019) and integrating the $NO_3$ production term from the beginning of the night ($t_0$) to the time of the measurement ($t$) according to Eq. (2).

$$[NO_3]_{int} = \int_{t_0}^{t} k_1[O_3][NO_2](t) \, dt \qquad (2)$$

In order to account for the pre-sunset production of $NO_3$ at high solar zenith angles where $N_2O_5$ could already be detected, $t_0$ was defined as the point in time at which $J_{NO_3}$ was below 0.017 s$^{-1}$ (about 10 % of maximum value during day). This was

typically 30-50 minutes prior to sunset. All data points before sunset were however excluded from the analysis due to the increased uncertainty in the reaction time. The $NO_2$ mixing ratio at the beginning of the night, $[NO_2]_0$, was derived from the measured $NO_2$ mixing ratio at time $t$ via Eq. (3) by assuming that $NO_2$ had been consumed by reaction with $O_3$ but the $O_3$ mixing ratio did not change significantly. Consequently, the amount of $NO_3$ produced along the air mass trajectory is equal to the difference between calculated $[NO_2]_0$ and measured $[NO_2]$ (t).

$$[NO_2](t) = [NO_2]_0 \, e^{-k_1[O_3]t} \qquad (3)$$



The ClNO$_2$ production efficiency ε can be determined by inserting the integrated NO$_3$ production over the course of the night and the measured ClNO$_2$ mixing ratio (assuming no losses) into Eq. (4).

$$\varepsilon = \frac{[\text{ClNO}_2]}{[\text{NO}_3]_{\text{int}}} \qquad (4)$$

To account for fresh emissions of NO (e.g. by passing ships), the reaction time $t'$ was calculated from Eq. (5) according to McDuffie et al. (2018a):

$$t' = -(k_1[\text{O}_3]s) \, ln\left(\frac{[\text{NO}_2]}{[\text{NO}_y]}\right) \qquad (5)$$

where $s$ represents the number of NO$_2$ molecules required to make NO$_y$ and is 1 when NO$_3$ reacts directly with VOCs and 2 when NO$_3$ reacts with NO$_2$ to form N$_2$O$_5$, which subsequently hydrolyses to HNO$_3$. As discussed later, the direct NO$_3$ losses are dominant throughout the campaign compared to the heterogeneous N$_2$O$_5$ production, so to a good approximation, $s = 1$. As discussed by McDuffie et al. (2018a) inherent to the use of this expression is the assumption that NO$_y$ is conserved during the night; any losses of NO$_y$ (e.g. via deposition of HNO$_3$) lead to an underestimation of the true reaction time. As the calculated, night-time air mass age depends on the ratio between [NO$_2$] and [NO$_y$], the calculation breaks down whenever a fresh NO emission (e.g. from a nearby ship) is injected into an air-mass and unreacted NO is still present. To avoid this, we only analyse ClNO$_2$ data when NO is below the detection limit. In addition, we only consider data points where the calculated age of the air mass is equal to or exceeds the time elapsed since sunset as these air masses are unlikely to have been impacted by recent emissions. As the loss of NO$_z$ via deposition will result in an air mass age that is shorter than the true one, we relax the criterion for equality of reaction times by also including calculated air mass ages that are up to 25 % shorter (i.e. $t' \geq 0.75 \, (t - t_0)$). Assuming a deposition velocity of 1 cm s$^{-1}$ (McDuffie et al., 2018a) for HNO$_3$ (which is the dominant component of NO$_z$ during AQABA) and a boundary layer height of 1000 m results in a lifetime of HNO$_3$ with respect to deposition of about 28 h, i.e. within one night (typical duration of maximum 12 h) less than 35 % of the HNO$_3$ will be lost. The reduced dataset provides 1742 data points with a median value of ε = 2.7 % for the whole campaign. The data reduction is described in more detail in the supplement, where the sensitivity of ε to these limitations and additional constraints are discussed. Here, we emphasise that even when limiting the dataset to ClNO$_2$ mixing ratios exceeding 100 pptv, the ClNO$_2$ production efficiency still remains relatively low (ε = 6.4 %, Fig. S9). Although the median values in the box plot in Fig. 6 would be modified (Fig. S10), the relative differences between the regions and especially the low ε observed over the Arabian Gulf persist.

### 3.3 Temporal and regional variability in ε

To compare the efficiency of ClNO$_2$ formation in different regions, a median value for ε was derived for each individual night with the results illustrated in Fig. 5 in which the size of the circles (1$^{st}$ leg) and stars (2$^{nd}$ leg) scales with the median NO$_3$ production rate. Despite high NO$_3$ production rates (high O$_3$ and NO$_2$ levels), the lowest values of ε were observed over





the Arabian Gulf, whereas elevated values of ε were found e.g. over the Arabian Sea, where $NO_3$ production was lowest. Reasons for this are discussed in the following section.

Fig. 6 displays box-plots of ε for each region, calculated from between 41 and 546 datapoints per region. The median $ClNO_2$ production efficiency ε displays large night-to-night variability and interregional variability with the highest value found over the Gulf of Aden and the Arabian Sea (median = 4.7 %) and the lowest value found over the Arabian Gulf (median = 0.8 %). Median values of ε (in %) derived for the Mediterranean Sea, the Suez Canal, the Red Sea and the Gulf of Oman were 2.9, 2.7, 2.1 and 2.0. In the following, we examine the factors that cause the generally low efficiency in $ClNO_2$ production and also the regional variability in ε.

### 3.4 Factors influencing the $ClNO_2$ production efficiency

The uptake of $N_2O_5$ to aerosol particles can proceed via hydrolysis to $HNO_3$ as well as formation of $ClNO_2$, with yield $f$. Assuming no night-time losses, the concentration of $ClNO_2$ is given by Eq. (6). The overall $ClNO_2$ production efficiency ε as derived in Sect. 3.2 is dependent on the relative rates of direct $NO_3$ loss ($k_{dir}$, Eq. 7) and indirect $NO_3$ loss ($k_{het}$, Eq. 8) where $A$ is the particle surface area concentration and $\bar{c}$ is the mean molecular velocity of $N_2O_5$ (24,400 ± 160 cm s$^{-1}$ during AQABA). During night-time, $k_{dir}$ is determined by the $NO_3$ reactivity towards VOCs and NO, $k_{het}$ by the rate of heterogeneous uptake of $N_2O_5$ to aerosol particles.

$$[ClNO_2] = [NO_3]_{int} \times \varepsilon = [NO_3]_{int} \times \left(\frac{k_{het}}{k_{het}+k_{dir}}\right) \times f \tag{6}$$

$$k_{dir} = \sum_i (k_{VOC})_i [VOC]_i \tag{7}$$

$$k_{het} = \gamma \left(\frac{A\bar{c}}{4}\right) k_{eq} [NO_2] \tag{8}$$

If no particulate chloride is available, $f$ is zero and two $NO_3^-$ ions are produced according to Reaction (R1) whereas if the particulate chloride concentration is large, $f$ approaches unity and one $NO_3^-$ anion plus one $ClNO_2$ molecule are formed. As particulate nitrate ($NO_3^-$) can leave the particle as $HNO_3$ we can write:

$$f = 2 \left(\frac{p[NO_3^-]+p[HNO_3]}{p[ClNO_2]} + 1\right)^{-1} \tag{9}$$

where $p$ signifies a production rate. Equation (9) assumes that $N_2O_5$ uptake is the sole mechanism for the night-time production of $HNO_3$. In principal, as demonstrated by Phillips et al. (2016), $f$ can be derived from field data using measured production rates (or concentrations) of inorganic nitrate ($NO_3^-$ + $HNO_3$) and $ClNO_2$ according to Eq. (9). For AQABA, we derived the mixing ratio of total inorganic nitrate from measurements of $NO_z$ (corrected for $ClNO_2$ and $N_2O_5$) assuming that, for this marine environment the contribution of organic nitrate in both gas- and particle phases is small compared to inorganic nitrate. $f$ could be derived using Eq. (9) whenever there was a significant correlation (over a period of several hours) between $ClNO_2$ and inorganic nitrate, as illustrated in Fig. 7 for data obtained in the Gulf of Oman for which $f = 0.60 \pm 0.04$.





The spatially and temporally variable sources of pollution during AQABA meant that requirement of a homogeneous fetch over periods of hours was rarely fulfilled and only a handful of values for $f$ could be derived this way. Further values derived are $0.42 \pm 0.06$ (Red Sea, 15.07.), $0.84 \pm 0.09$ (Gulf of Oman, 24./25.07.) and $0.65 \pm 0.05$ (Mediterranean Sea, 29.08.), indicating that the values of $f$ were generally large whenever $ClNO_2$ was observed.

$f$ can also be calculated (Eq. 10) if the relative concentrations of particulate chloride and water are known (Behnke et al., 1997; Bertram and Thornton, 2009):

$$f = \left( \frac{k_{Cl^-} \cdot ([Cl^-])}{k_{Cl^-}[Cl^-] + k_{H2O}[H_2O]} \right) = \left( 1 + \frac{[H_2O]}{\frac{k_{Cl^-}}{k_{H2O}}[Cl^-]} \right)^{-1} \tag{10}$$

where $\left( \frac{k_{Cl^-}}{k_{H2O}} \right) = 450$ is the ratio of rate constants (Ammann et al., 2013) for reaction of $NO_2^+$ (formed along with $NO_3^-$ upon dissociation of $N_2O_5$ in the aqueous-phase) with either $Cl^-$ or $H_2O$. The aerosol liquid water content $[H_2O]$ and chloride ion

concentration $[Cl^-]$ were calculated using the E-AIM model (http://www.aim.env.uea.ac.uk/aim/model4/model4a.php, (Clegg et al., 1998; Friese and Ebel, 2010) using the ambient temperature and relative humidity along with $PM_1$ nitrate, sulphate, ammonium and chloride mass concentrations ($\mu g\ m^{-3}$) as reported by the AMS.

From the median aerosol composition ($PM_1$) in the seven different regions we calculated median values of $f$, which are shown in Table 2. The values of $f$ obtained via Eq. (10) were variable between regions, with medians of 0.53 in the

Mediterranean Sea, 0.90 in the Suez Canal, 0.86 in the Red Sea, 0.76 in the Gulf of Aden, 0.87 in the Indian Ocean, 0.50 in the Gulf of Oman and 0.17 in the Arabian Gulf. To put these numbers in context, a value of $f = 0.9$ corresponds to a $\approx 1.1$ mol $L^{-1}$ $Cl^-$ solution. MARGA measurements of $Cl^-$ and $Na^+$ (total suspended particles, TSP) also indicated sea salt concentrations up to 20 $\mu g\ m^{-3}$ in the coarse mode over the Arabian Sea. A comparison of the $[Cl^-]$ (TSP) from the MARGA (which detects NaCl as well as $NH_4Cl$) with the (largely) non-refractory $[Cl^-]$ reported by the AMS reveals a strong co-

variance. The AMS concentrations ($PM_1$) were on average $\approx 1\%$ of those reported by the MARGA (TSP) with the correlation between them indicating that the AMS chloride ($PM_1$) is mainly due to sea-salt rather than $NH_4Cl$. If we assume that $\approx 10\%$ of the sea salt mass (TSP) is associated with the fine mode ($PM_1$) as previously derived (Sommariva et al., 2018), we can use the MARGA (TSP) $[Cl^-]$ to estimate that the true $PM_1$ $[Cl^-]$ would be about an order of magnitude higher than measured by the AMS. Under the assumption that this is true, $f$ is $> 0.67$ in all seven regions implying that a lack of $Cl^-$ is not

the reason for low values of $\varepsilon$, as may be expected for a marine environment.

Previous derivations of $f$ in a marine environment (Texas coast) yield values between 0.1 and 0.65 (Osthoff et al., 2008) whereas even larger values (up to 0.9) have been reported for inland sites impacted by anthropogenic emissions (Young et al., 2013) or by long-range transport of sea-salt $(0.035 < f < 1)$ (Phillips et al., 2016). A median value of 0.138 $(0.003 < f < 1)$ was derived for airborne measurements in a coastal region during winter (McDuffie et al., 2018b).

The uptake coefficient, $\gamma$, can be estimated using the parameterisation in Eq. (11) (Bertram and Thornton, 2009):



$$\gamma = Bk \times \left(1 - \left(\frac{k_2[\text{H}_2\text{O}(l)]}{k_{-1}[\text{NO}_3^-]}\right) + 1 + \left(\frac{k_4[\text{Cl}^-]}{k_{-1}[\text{NO}_3^-]}\right)^{-1}\right) \qquad (11)$$

where $Bk = 3.2 \times 10^{-8}$ s, $k = 1.15 \times 10^6 - 1.15 \times 10^6 \exp(-0.13 [\text{H}_2\text{O}(l)])$ s$^{-1}$, $k_2/k_{-1} = 0.06$ and $k_4/k_{-1} = 29$. Using AMS data for [NO$_3^-$] and [Cl$^-$] we derive $\gamma = 0.033 \pm 0.003$ where the standard deviation encompasses the weak inter-regional variation (see Table 2). This value is consistent with $\gamma = 0.03 \pm 0.02$ reported by Aldener et al. (2006) for a polluted marine
environment. The large values of $\gamma$ reflect the low PM$_1$ particulate nitrate concentrations observed during AQABA where the high temperatures favour the partitioning of particulate nitrate and HNO$_3$(g) to the gas-phase. A suppression of $\gamma$ through the presence of organics in the particle phase has been reported (Bertram et al., 2009), though the low (generally < 2) organic-to-sulphate ratio in particles observed during AQABA suggest that this is not likely to be important for the present analysis.

The campaign averaged, fractional contribution of coarse mode particles (PM$_{10}$ - PM$_1$) to the total particle surface area
concentration ($A$) was only 14 ± 14 % so that the uncertainty incurred when using the PM$_1$ particle surface area concentration to derive the heterogeneous NO$_3$ loss rate ($k_{\text{het}}$, Table 2) from Eq. (8) is negligible. However, during two periods (of 2–5 days duration) when the ship was sailing through the Gulf of Aden / Arabian Sea / Southern Red Sea and air masses originated from the deserts of Eritrea, Djibouti and Ethiopia, the contribution from the coarse mode particles to the aerosol surface area increased to about 60 % mainly due to dust (and sea-salt). The relative contribution of dust and sea-salt
to the coarse mode was estimated using MARGA measurements of Ca$^{2+}$ and Na$^+$ in TSP. The dust loading was calculated by assuming that the dust aerosols of Saharan origin consist of 10 % calcium (Molinaroli et al., 1993). Freshly generated dust particles do not contain chloride and the uptake of N$_2$O$_5$ to them ($\gamma = 0.02 \pm 0.01$ for Saharan dust, Tang et al. (2012)) does not result in ClNO$_2$ formation ($f = 0$) but contributes to $k_{\text{het}}$ through the additional aerosol surface area, thus lowering the ClNO$_2$ production efficiency. In contrast, the uptake of N$_2$O$_5$ to coarse mode sea-salt particles has a ClNO$_2$ yield $f$ close to
unity (Ammann et al., 2013), and can therefore enhance formation of ClNO$_2$. Throughout the whole campaign, the contribution of dust to the uptake of N$_2$O$_5$ to coarse mode particles was much larger than that of sea-salt (on average 13 ± 10 % with a range of 0–40 %). A close examination of the periods (14.–19.07. and 15.–17.08.) strongly influenced by dust particles did not reveal any (anti)correlation between elevated dust concentration and ClNO$_2$ mixing ratios. This is also reflected by the relatively high median value of $\varepsilon$ over the Gulf of Aden where the highest concentrations of coarse mode
particles were observed.

Using a campaign average of $\varepsilon = 2.7$ % and a maximum value of $f = 1$ we can show that, with 97.3 %, the direct (gas-phase) loss of NO$_3$ ($k_{\text{dir}}$) is much more important than indirect losses via N$_2$O$_5$ uptake ($k_{\text{het}}$), which contribute the remaining 2.7 %. Assuming a very conservative estimate of $f = 0.5$ would still result in contribution of 94.6 respectively 5.4 %. To put this into context, over the Gulf of Oman we calculated a median value of $k_{\text{het}} = 6.5 \times 10^{-4}$ s$^{-1}$ (Table 2) which would result (assuming
$\varepsilon = 2.0$ % and $f = 0.5$) in a direct NO$_3$ loss term towards VOCs of $k_{\text{dir}} = 1.6 \times 10^{-2}$ s$^{-1}$. The relatively small contribution of the heterogeneous loss term is readily explained by high mean night-time temperatures of 25–35 °C during AQABA, which favour the existence of NO$_3$ rather than N$_2$O$_5$. To illustrate this, we calculate that a nocturnal temperature of 20 °C (rather

than 30 °C) would increase the contribution of $k_{het}$ to 21% of the total $NO_3$ loss and lead to 3 times higher $ClNO_2$ mixing ratios assuming that the rate constants for reaction between $NO_3$ and reactive trace gases are not strongly temperature-dependent. A further reduction in temperature to 10 °C would lead to equality in $k_{het}$ and $k_{dir}$ and result in a factor of seven more $ClNO_2$ than observed.

The direct (gas-phase) loss rate of $NO_3$ can also be calculated from Eq. (7) if the concentrations of all VOCs contributing to its reactivity are known. Analysis of the steady-state-lifetimes of $NO_3$ during AQABA  however frequently revealed that much of the $NO_3$ reactivity could not be attributed to measured trace-gases including $NO$, $CH_3SCH_3$, isoprene and various VOCs. A detailed analysis of the $NO_3$ lifetime and the role of VOCS will be presented in a separate publication.

### 3.5 Comparison with literature

In the following, we compare the generally low values of ε derived during AQABA with previous determinations. Mielke et al. (2013) report a yield of $ClNO_2$ relative to the total amount of $NO_3$ formed at night of 0.7 % to 62 % with a median of 12% in the polluted coastal boundary layer in Pasadena, California. In contrast, $ClNO_2$ production efficiencies derived for the urban boundary layer of Calgary, Canada (Mielke et al., 2016), were significantly lower, ranging from 0.1 % to 4.5 % (10[th] and 90[th] percentiles, median 1.0 %). Osthoff et al. (2018) report very low efficiencies with a median of 0.17 % and a

maximum of 5.4 % for the Lower Fraser Valley of British Columbia, Canada, potentially due to a lack of available aerosol chloride. For AQABA we derive a region-dependent median efficiency of 1–5 % with a campaign median of 2.8 %, despite similar conditions to Mielke et al. (2013), i.e. mostly polluted marine environment. The difference can be attributed to exceptionally high nocturnal temperatures during AQABA with a median of 25–35 °C for different regions, shifting the equilibrium from $N_2O_5$ towards $NO_3$ and favouring direct $NO_3$ losses. For comparison, daily minimum temperatures during

the study reported by Mielke et al. (2013) were 10–20 °C (Ryerson et al., 2013) whereas the lowest temperature measured during the whole AQABA cruise was 22 °C. Based on the $ClNO_2$ dataset reported by Phillips et al. (2012) for continental Germany, where $ClNO_2$ mixing ratios up to 800 pptv were reported, we calculate values of ε that range from 0.4 % to 12.3 % (10[th] and 90[th] percentiles) with a median of 2.6 %. On nights where $ClNO_2$ mixing ratios above 100 pptv were observed, range and median increase to 5.0–24.1 % and 10.6 % respectively. Compared to AQABA the yield per $NO_3$ molecule formed

on nights where $ClNO_2$ was present at levels > 100 pptv is about a factor of 2 higher for this dataset, again most likely a result of the lower nocturnal temperatures.

### 3.6 Cl atom generation from ClNO₂ and HCl

In this section we assess the role of two gas-phase chlorine reservoirs, $ClNO_2$ and HCl as sources of Cl atoms during AQABA. Compared to the complex route to $ClNO_2$ formation described above, the formation of HCl in the polluted marine

environment can be traced back to its displacement from sea-salt particles by stronger acids, such as $H_2SO_4$ and $HNO_3$ (Keene et al., 1999).


The high emission rates of $NO_x$ and $SO_2$ by ship traffic resulted in enhanced concentrations of both $NO_2$ and $SO_2$ (see Fig. S7) during parts of the AQABA campaign. Both $NO_2$ and $SO_2$ are oxidised via OH to form $HNO_3$ and $H_2SO_4$, both of which can be taken up by sea-salt containing aerosol releasing HCl. The release of HCl through acid displacement leads to a deficit

in particulate $Cl^-$ concentrations which can be expressed in terms of a chloride depletion factor (Eq. 12) where $[Na^+]$ and $[Cl^-]$ represent the concentrations in mol $m^{-3}$ and 1.174 is the molar ratio of $Cl^-$ to $Na^+$ found in sea-water (Zhuang et al., 1999).

$$Cl\ depletion\ (\%) = 100\ \times\ \frac{(1.174 \times [Na^+] - [Cl^-])}{1.174 \times [Na^+]} \tag{12}$$

In Fig. 8 we present a time series (Gulf of Oman / Arabian Gulf) in which significant differences in particulate $Na^+$ and $Cl^-$

concentrations coincide with high mixing ratios of $NO_2$, $SO_2$ and HCl (chloride depletion up to 90 %) indicating efficient HCl acid displacement by $HNO_3$ and $H_2SO_4$ in this region.

The instantaneous production rate of Cl atoms from the photolysis of $ClNO_2$, $P_{Cl}(ClNO_2)$, is given by the photolysis rate constant for $ClNO_2$ ($J_{ClNO2}$) and its concentration (Eq. 13) whereas the instantaneous Cl production rate from HCl, $P_{Cl}(HCl)$, requires knowledge of the OH concentration (Eq. 14) and the rate coefficient for reaction between OH and HCl ($k_{OH+HCl}$ =

$1.7 \times 10^{-12}$ exp(-230/T) $cm^3$ molecule$^{-1}$ s$^{-1}$, (Atkinson et al., 2007; IUPAC, 2019)).

$$P_{Cl}(ClNO_2) = J_{ClNO2}\ \times [ClNO_2] \tag{13}$$

$$P_{Cl}(HCl) = k_{OH+HCl}\ [OH][HCl] \tag{14}$$

In the following analysis, we focus on two consecutive nights in the Gulf of Oman region (Fig. 9) where we observed a monotonous increase of $ClNO_2$ mixing ratios up to $\approx$ 300 pptv during the second half of the night followed by a decrease

over a 4 hour period starting at sunrise (upper panel). The corresponding Cl production rate from $ClNO_2$ photolysis reaches a maximum of $0.8 \times 10^6$ molecule $cm^{-3}$ s$^{-1}$ on the first night and $0.7 \times 10^6$ molecule $cm^{-3}$ s$^{-1}$ on the second night. To place this in context, we also make a rough estimate (lower limit) to the rate of OH radical production ($P_{OH}(O_3)$) from the photolysis of $O_3$ in the presence of $H_2O$ (Eq. 15).

$$P_{OH}(O_3) = \frac{2\,J_{O1D}\,[O_3] + k_{H2O}\,[H_2O]}{k_{H2O}\,[H_2O] + k_{N2}\,[N_2] + k_{O2}\,[O_2]} \tag{15}$$

Where $J_{O1D}$ is the photolysis rate constant for $O_3$, $k_{H2O} = 2.4 \times 10^{-11}$ $cm^3$ molecule$^{-1}$ s$^{-1}$, $k_{N2} = 2.15 \times 10^{-11}$ exp(110/T) $cm^3$ molecule$^{-1}$ s$^{-1}$ and $k_{O2} = 3.2 \times 10^{-11}$ exp(67/T) $cm^3$ molecule$^{-1}$ s$^{-1}$ (IUPAC, 2019) refer to reactions of $O(^1D)$ with $H_2O$, $N_2$ and $O_2$, respectively.

Although the maximum, midday OH production rates from $O_3$ photolysis (~$1 \times 10^7$ OH molecule $cm^{-3}$ s$^{-1}$) are about an order of magnitude higher than Cl atom production rates, during the first ~2 hours after sunrise, $P_{Cl}$ and $P_{OH}$ are roughly equal for

this particular case-study.



In order to examine the regional dependence of Cl formation we calculated the ClNO$_2$ production rate from the regional median values of ε, the NO$_3$ production rate and the length of the night as described previously. As ClNO$_2$ is completely photolysed to Cl during daytime, the total Cl-production from ClNO$_2$ photolysis (ΣCl$_{ClNO2}$) is equal to the median amount of ClNO$_2$ formed during the night. The total Cl-production from HCl + OH (ΣCl$_{HCl}$) was calculated by integrating the

production rate $P_{Cl}$(HCl) (based on [HCl] and [OH] measurements, Eq. 14) over the median diurnal profile. To calculate the contribution of O$_3$ photolysis to the OH production (ΣOH$_{O3}$), we integrated $P_{OH}$(O$_3$) (calculated via Eq. 15) over the median diel profile.

In Table 3 we summarise the average, total daytime production of OH and Cl in the seven regions. The median radical production over one diurnal cycle calculated for the Suez region is exemplified in Fig. 10, whereby we assume an initial 363

pptv of ClNO$_2$ at sunrise (Table 3), which is subsequently photolysed according to $J_{ClNO2}$. Considering the entire campaign, we conclude that Cl-atom formation via ClNO$_2$ photolysis and OH-initiated HCl oxidation are of similar magnitude, though their relative contributions show large regional variability. For example, Cl formation from OH + HCl (ΣCl$_{HCl}$) is a factor ~ 10 more important than ClNO$_2$ photolysis (ΣCl$_{ClNO2}$) over the Mediterranean Sea where the NO$_3$ (and thus ClNO$_2$) production rate was low owing to low NO$_x$ levels and where the OH concentrations were highest. This is consistent with Li

et al. (2019) who indicate the importance of HCl over the eastern Mediterranean Sea for which monthly average mixing ratios of 0.5–1.5 ppbv were predicted by a regional model. ΣCl$_{HCl}$ is also a factor ≈ 2 higher than ΣCl$_{ClNO2}$ over the Arabian Gulf where the ClNO$_2$ production efficiency, ε, was low. In all other regions ΣCl$_{ClNO2}$ was about equal to ΣCl$_{HCl}$ or higher by factors between ≈ 1 and 4. This in in broad agreement with Riedel et al. (2012a) who report roughly equal importance of ClNO$_2$ and HCl as chlorine atom sources in the polluted marine boundary layer of the Los Angeles region. They also report a

maximum midday Cl production rate from ClNO$_2$ photolysis of 0.6 x 10$^6$ molecule cm$^{-3}$ s$^{-1}$, which is similar to the production rates we obtained over the Gulf of Oman (Fig. 9).

When comparing the total number of chlorine atoms generated during the day (ΣCl$_{total}$ = ΣCl$_{ClNO2}$ + ΣCl$_{HCl}$) with the total number of OH formed from O$_3$ photolysis (ΣOH$_{O3}$) over the same period, we find the expected domination of OH for all regions. The largest contribution of Cl to the total radical production (4 %) was observed over the Suez Canal where the Cl

production was highest (about 32 % from HCl and 68 % from ClNO$_2$). The lowest ratio of Cl-to-OH production was observed over the Arabian Sea, reflecting the low levels of NO$_x$ and very low rates of NO$_3$ generation. Although only (at maximum) ≈ 4 % of the radicals generated are Cl atoms, they react more rapidly than OH with some classes of hydrocarbons, especially saturated hydrocarbons and small oxygenates. For example, the relative, room temperature rate coefficients ($k_{Cl}$ / $k_{OH}$) are 16, 61, 127 and 242 for reaction with CH$_4$, CH$_3$OH, C$_3$H$_8$ and C$_2$H$_6$, respectively (Atkinson et al.,

2004; IUPAC, 2019). The impact of chlorine atoms is thus mainly seen in the oxidation rates of unsaturated hydrocarbons, the relative concentrations of which may be modified according to the relative abundance of OH and Cl and the relative reaction rate constants. For the AQABA campaign, evidence for such effects has been reported by Bourtsoukidis et al.



(2019), and for the global scale Wang et al. (2019) conclude that oxidation by Cl atoms accounts for 1.0 % of methane loss with larger impacts on ethane (20 %), propane (14 %), and methanol (4 %).

## 4 Summary and conclusion

The AQABA campaign, which took place in summer 2017 along the sea route from southern France to Kuwait provided the first $ClNO_2$ measurements in the marine boundary layer of the Mediterranean Sea, the Suez Canal, the Red Sea, the Gulf of Aden, the Arabian Sea, the Gulf of Oman and the Arabian Gulf. Along the ship track we observed a large variability in $ClNO_2$ mixing ratios with nocturnal maxima ranging from below the detection limit over the Indian Ocean to a few hundred pptv over the Gulf of Oman, the northern part of the Red Sea, the Gulf of Suez and the Mediterranean Sea close to Sicily / Italy with a campaign maximum of $\approx$ 600 pptv observed over the Gulf of Suez.

The overall $ClNO_2$ production efficiency, i.e. the yield of $ClNO_2$ per $NO_3$ molecule formed in the reaction of $NO_2$ with $O_3$, was generally low (median of 2.7 % for the whole campaign) and highly variable within individual nights and between different regions with values (in percent) of 2.9, 2.7, 2.1, 4.7, 4.7, 2.0, and 0.8 over the Mediterranean Sea, the Suez Canal, the Red Sea, the Gulf of Aden, the Arabian Sea, the Gulf of Oman and the Arabian Gulf, respectively. The relatively low $ClNO_2$ production efficiency compared to previous measurements in the polluted marine boundary layer or at continental sites was attributed to high nocturnal temperatures during AQABA (25–35 °C), which significantly shifted the equilibrium between $NO_3$ and $N_2O_5$ towards $NO_3$ and lowered the importance of $N_2O_5$ uptake to particles relative to direct $NO_3$ losses. The low $ClNO_2$ production efficiency in the Arabian Gulf (< 1 %) results from a combination of high temperatures, enhanced $NO_3$ reactivity and lowered chloride availability. The photolysis of $ClNO_2$ was found to represent an important source of chlorine radicals in the early morning in areas where efficient night-time production was observed, and was augmented (and sometimes exceeded) by Cl atoms formation from the reaction of OH with HCl, especially in areas where ppbv levels of HCl were observed such as the Mediterranean Sea or the Arabian Gulf.

Although the amount of Cl atoms generated were found to be a factor 25 to 300 less than the amount of OH molecules generated from $O_3$ photolysis, the high rate coefficients ratio for Cl compared to OH reactions towards some hydrocarbons imply that Cl may enhance hydrocarbon oxidation, especially in the early morning.

## Data availability

The datasets presented in this field study are archived and distributed through the KEEPER service of the Max Planck Digital Library (https://keeper.mpdl.mpg.de). They will be available from August 2019 subsequent to agreeing to the AQABA data protocol.



## Author contributions

Philipp Eger performed the CI-QMS measurements of $ClNO_2$, $HCl$ and $SO_2$ during the AQABA campaign, evaluated the field data and wrote the manuscript. John Crowley operated the CI-QMS and the CRDS-instruments during parts of the first leg and supervised the study. $NO_2$ and $N_2O_5$ data were provided by Justin Shenolikar. $NO_x$, $NO_y$, and $NO_z$ data were provided by Nils Friedrich. J-values were measured by Jan Schuladen. Ivan Tadic and Horst Fischer contributed the NO and $NO_2$ datasets. AMS, OPC and FMPS measurements and analysis were performed by James Brooks, Eoghan Darbyshire, Friederike Fachinger and Frank Drewnick. MARGA data was provided by Michael Pikridas and Jean Sciare. Roland Rohloff, Sebastian Tauer, Monica Martinez and Hartwig Harder provided the OH dataset. Jos Lelieveld designed the AQABA campaign, all authors contributed to the manuscript.

## Competing interests

The authors declare that they have no conflict of interest.

## Acknowledgements

We acknowledge the co-operation with the Cyprus Institute (CyI), the King Abdullah University of Science and Technology (KAUST) and the Kuwait Institute for Scientific Research (KISR). We are grateful for the support of Hays Ships Ltd, the captain and his crew on-board the *Kommandor Iona*. We thank Marcel Dorf and Claus Koeppel for logistical support.





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





**Table 1:** Measured trace gases and other parameters (5 min, night-time only) for the different regions indicated in Fig. 2.

| Parameter[1] | Med. Sea[2] | Suez | Red Sea | Aden | Arab. Sea | Oman | Arab. Gulf |
|---|---|---|---|---|---|---|---|
| **ClNO$_2$ [pptv]** | N = 751 | N = 376 | N = 1247 | N = 451 | N = 918 | N = 475 | N = 500 |
| Mean ± STD[1] | 20 ± 39 | 75 ± 101 | 47 ± 55 | 41 ± 56 | 7 ± 8 | 67 ± 71 | 21 ± 18 |
| median | 9 | 36 | 22 | 19 | 6 | 39 | 18 |
| range | LOD–439 | LOD–586 | LOD–480 | LOD–379 | LOD–56 | LOD–376 | LOD–126 |
| **HCl [ppbv]** | N = 749 | N = 375 | N = 1245 | N = 449 | N = 918 | N = 473 | N = 498 |
| Mean ± STD | 0.68 ± 0.52 | 0.94 ± 0.63 | 0.81 ± 0.36 | 0.28 ± 0.46 | 0.00 ± 0.13 | 0.86 ± 0.70 | 1.20 ± 0.99 |
| median | 0.71 | 0.98 | 0.76 | 0.12 | LOD | 0.71 | 0.92 |
| range | LOD–3.09 | LOD–3.54 | LOD–2.58 | LOD–1.98 | LOD–0.38 | LOD–3.23 | LOD–4.49 |
| **SO$_2$ [ppbv]** | N = 751 | N = 376 | N = 1247 | N = 451 | N = 918 | N = 475 | N = 500 |
| Mean ± STD | 0.64 ± 0.65 | 2.84 ± 3.87 | 0.81 ± 0.77 | 1.26 ± 2.89 | 0.18 ± 0.65 | 2.38 ± 2.26 | 3.71 ± 2.01 |
| median | 0.46 | 1.57 | 0.66 | 0.55 | 0.07 | 1.39 | 3.25 |
| range | 0.06–7.24 | 0.06–34.98 | 0.04–9.75 | LOD–30.34 | LOD–12.33 | 0.12–12.34 | 0.80–14.85 |
| **O$_3$ [ppbv]** | N = 751 | N = 376 | N = 1247 | N = 451 | N = 800 | N = 475 | N = 500 |
| Mean ± STD | 61.6 ± 10.2 | 45.4 ± 16.1 | 58.1 ± 13.8 | 28.0 ± 10.9 | 24.4 ± 3.4 | 36.0 ± 16.4 | 83.9 ± 36.2 |
| median | 63.4 | 46.3 | 59.6 | 25.7 | 25.6 | 30.0 | 69.8 |
| range | 37.8–82.8 | 1.7–88.1 | 10.6–90.3 | 3.1–55.3 | 1.3–29.6 | 10.2–83.0 | 24.2–163.3 |
| **NO$_2$ [ppbv]** | N = 547 | N = 288 | N = 1217 | N = 449 | N = 898 | N = 470 | N = 500 |
| Mean ± STD | 0.80 ± 1.57 | 7.37 ± 7.56 | 1.97 ± 3.50 | 2.94 ± 4.11 | 0.63 ± 1.57 | 4.57 ± 3.93 | 1.59 ± 2.34 |
| median | 0.21 | 5.27 | 0.97 | 1.49 | 0.21 | 3.59 | 0.80 |
| range | 0.01–14.45 | 0.24–44.03 | 0.01–56.81 | 0.07–25.87 | 0.01–20.88 | 0.12–30.38 | 0.16–23.22 |
| **$p$NO$_3$ [pptv s$^{-1}$]** | N = 547 | N = 288 | N = 1138 | N = 449 | N = 783 | N = 470 | N = 500 |
| Mean ± STD | 0.04 ± 0.07 | 0.27 ± 0.21 | 0.10 ± 0.13 | 0.06 ± 0.05 | 0.01 ± 0.02 | 0.16 ± 0.14 | 0.12 ± 0.13 |
| median | 0.01 | 0.23 | 0.05 | 0.04 | 0.004 | 0.13 | 0.08 |
| range | 0.00–0.82 | 0.01–1.01 | 0.00–1.23 | 0.00–0.25 | 0.00–0.18 | 0.00–0.91 | 0.01–1.19 |
| **$T$ [°C]** | N = 751 | N = 376 | N = 1247 | N = 451 | N = 918 | N = 475 | N = 500 |
| Mean ± STD | 26.7 ± 0.6 | 27.2 ± 1.7 | 31.3 ± 1.7 | 31.3 ± 1.1 | 25.4 ± 0.9 | 32.7 ± 1.3 | 34.4 ± 0.6 |
| median | 26.8 | 27.5 | 31.8 | 31.1 | 25.5 | 33.0 | 34.4 |
| range | 25.5–29.1 | 21.6–31.7 | 27.7–34.1 | 29.3–35.6 | 23.5–31.2 | 29.6–35.8 | 33.2–36.4 |
| **RH [%]** | N = 751 | N = 376 | N = 1247 | N = 451 | N = 918 | N = 475 | N = 500 |
| Mean ± STD | 79.0 ± 5.6 | 72.4 ± 9.1 | 73.2 ± 6.3 | 74.6 ± 10.2 | 88.8 ± 2.9 | 79.8 ± 8.9 | 76.5 ± 7.9 |
| median | 79.5 | 71.6 | 72.7 | 75.6 | 89.1 | 81.5 | 77.8 |
| range | 63.2–89.3 | 40.6–93.2 | 58.7–88.4 | 27.2–90.0 | 79.3–94.7 | 51.9–95.1 | 53.3–91.6 |
| **$A$ [μm² cm$^{-3}$]** | N = 690 | N = 373 | N = 1101 | N = 440 | N = 899 | N = 222 | N = 393 |
| Mean ± STD | 202 ± 61 | 240 ± 200 | 243 ± 62 | 111 ± 64 | 81 ± 62 | 363 ± 317 | 1010 ± 600 |
| median | 178 | 166 | 257 | 107 | 77 | 275 | 809 |
| range | 114–466 | 90–1134 | 111–376 | 35–338 | 23–426 | 101–1244 | 179–2726 |

Notes: [1] Parameter: STD = standard deviation; RH = relative humidity; $A$ = ambient PM$_1$ particle surface area concentration (see section 2.5); $p$ denotes a production rate; $N$ = number of data points; LOD = Limit of detection (see Sect. 2.2). [2] Regions: Mediterranean Sea (Med. Sea): 30.06.–01.07. and 24.08.–30.08. Suez Canal and Gulf of Suez (Suez): 02.07.–03.07. and 22.08.–23.08. Red Sea: 03.07.–15.07. and 17.08.–21.08. Gulf of Aden (Aden): 16.07.–18.07. and 15.08.–16.08. Arabian Sea (Arab. Sea): 19.07.–23.07. and 07.08.–14.08. Gulf of Oman (Oman): 24.07.–27.07. and 05.08.–06.08. Arabian Gulf (Arab. Gulf): 28.07.–04.08.



**Table 2:** Regional variability in ε, ClNO$_2$ yield ($f$), N$_2$O$_5$ uptake coefficient ($\gamma$) and heterogeneous NO$_3$ loss rate ($k_{het}$).

| Region | ε [%] | $f$ | $\gamma$[1] | $k_{het}$ [10$^{-5}$ s$^{-1}$] |
|---|---|---|---|---|
| Med. Sea | 2.9 | 0.53 | 0.034 | 4.4 |
| Suez | 2.7 | 0.90 | 0.031 | 69.7 |
| Red Sea | 2.1 | 0.86 | 0.031 | 12.9 |
| Gulf of Aden | 4.7 | 0.76 | 0.031 | 8.5 |
| Arab. Sea | 4.7 | 0.87 | 0.036 | 1.8 |
| Gulf of Oman | 2.0 | 0.50 | 0.033 | 64.8 |
| Arab. Gulf | 0.8 | 0.17 | 0.036 | 34.3 |

Notes: [1]Calculated from Eq. (11).

5  **Table 3:** Regional variability in ε, $p$NO$_3$ and OH and Cl radical production integrated over one diel cycle

| Region | ε [%] | $p$NO$_3$ [pptv s$^{-1}$] | ΣCl$_{ClNO2}$ [pptv] | ΣCl$_{HCl}$ [pptv] | ΣOH$_{O3}$ [pptv] | ΣCl$_{total}$/ΣOH [%][1] |
|---|---|---|---|---|---|---|
| Med. Sea | 2.9 | 0.012 | 18 | 234 | 12,364 | 2.0 |
| Suez | 2.7 | 0.231 | 363 | 170 | 13,216 | 4.0 |
| Red Sea | 2.1 | 0.053 | 48 | 53 | 12,411 | 0.8 |
| Gulf of Aden | 4.7 | 0.043 | 96 | 24 | 5,608 | 2.1 |
| Arab. Sea | 4.7 | 0.004 | 11 | 3 | 4,639 | 0.3 |
| Gulf of Oman | 2.0 | 0.130 | 155 | 127 | 12,649 | 2.2 |
| Arab. Gulf | 0.8 | 0.077 | 25 | 50 | 10,985 | 0.7 |

Notes: ΣCl$_{ClNO2}$ and ΣCl$_{HCl}$ are the integrated formation of Cl atoms from ClNO$_2$ and HCl respectively. ΣOH$_{O3}$ is the integrated formation of OH from O$_3$ photolysis. [1]ΣCl$_{total}$ = ΣCl$_{ClNO2}$ + ΣCl$_{HCl}$.



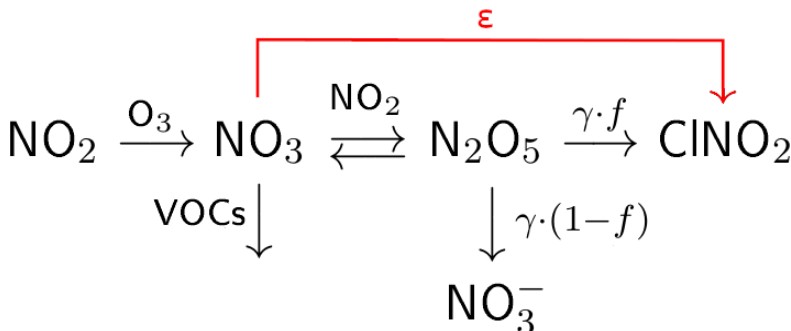

**Figure 1:** Simplified scheme of chemical reactions and parameters involved in the formation of $ClNO_2$. The $ClNO_2$ production efficiency per $NO_3$ formed is denoted as $\varepsilon$. The uptake coefficient is represented by $\gamma$ and the $ClNO_2$ yield (per $N_2O_5$ taken up) by $f$.



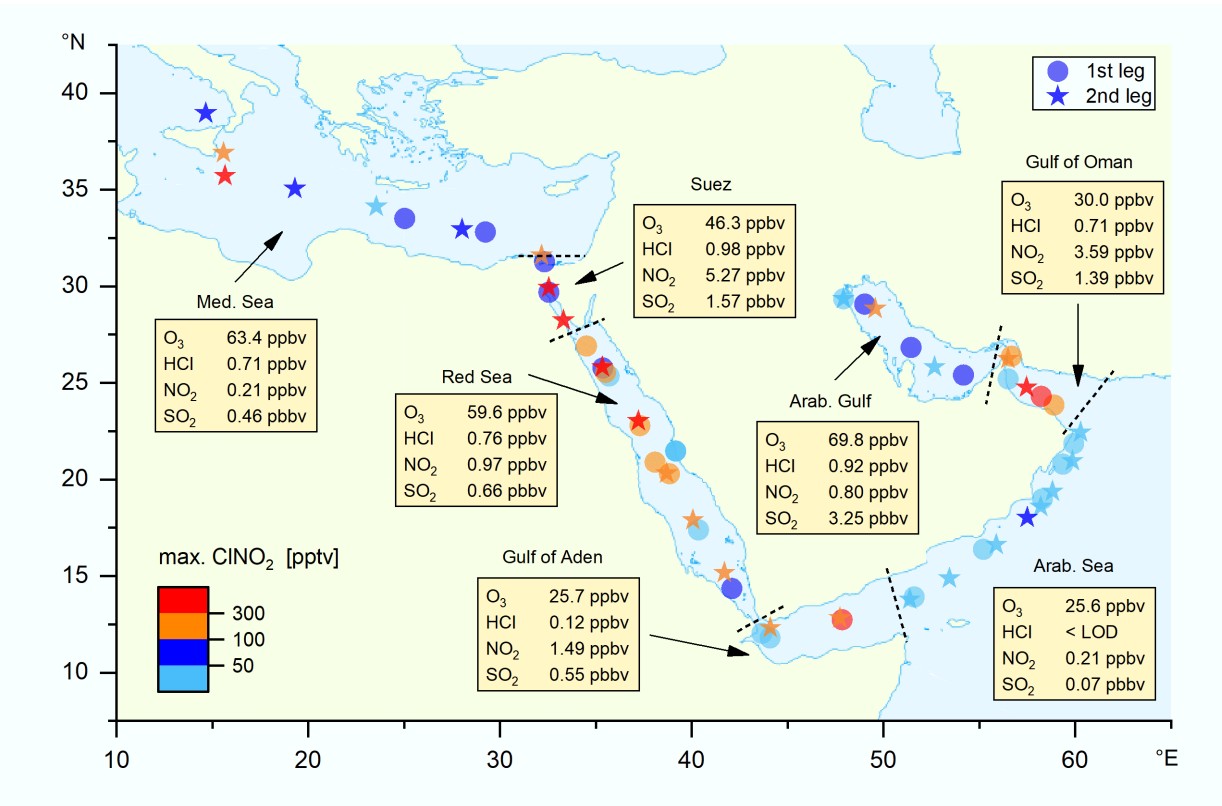

**Figure 2:** Map of maximum $ClNO_2$ mixing ratios on individual nights together with (median) night-time mixing ratios of $O_3$, HCl, $NO_2$ and $SO_2$ for different regions demarked by the black dashed lines. The circles and stars represent data obtained on the first and second legs, respectively.

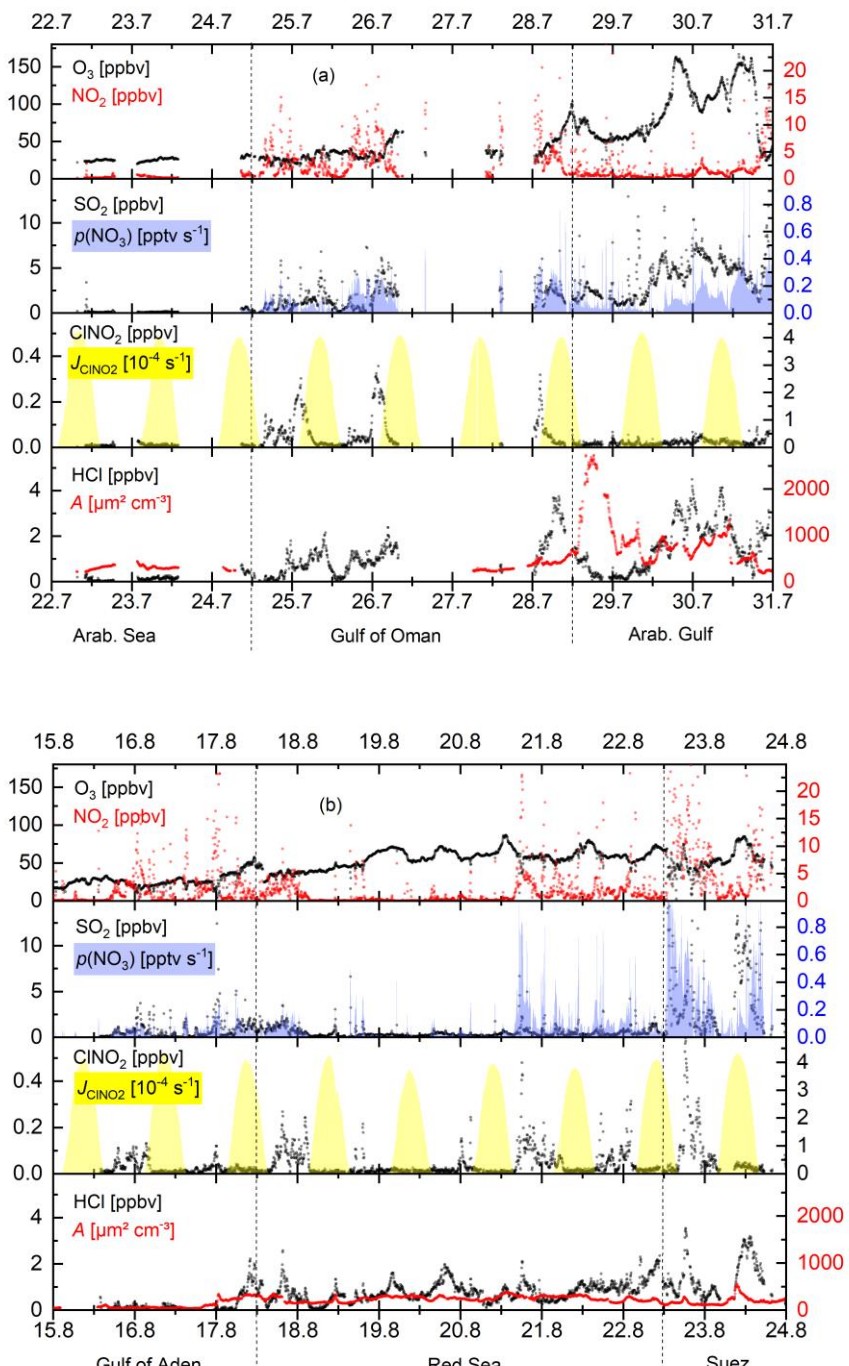

**Figure 3:** Time series of ClNO$_2$ and trace gases related to its production (NO$_2$ and O$_3$) as well as the NO$_3$ production rate, $p$(NO$_3$), ClNO$_2$ photolysis rate ($J_{ClNO2}$), PM$_1$ particle surface area concentration ($A$) and HCl mixing ratio in different regions (separated by the dashed lines).





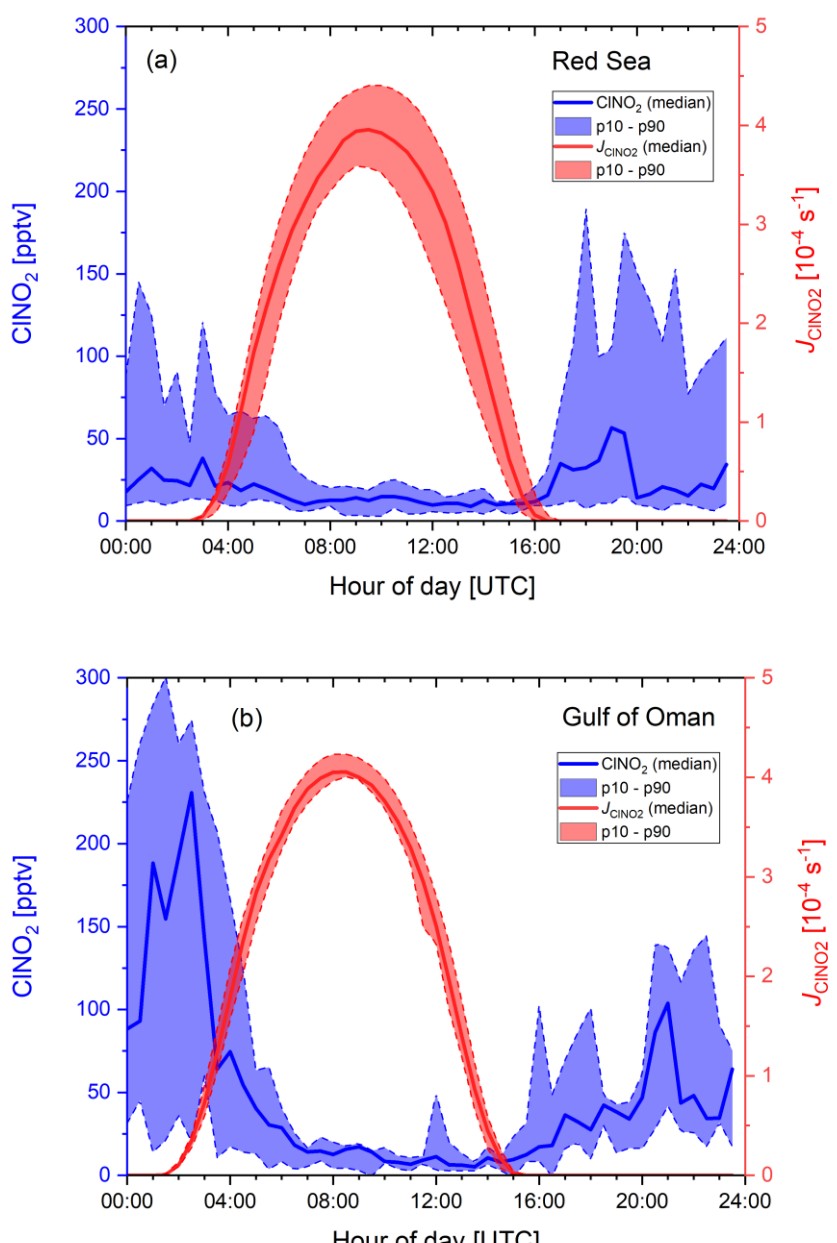

**Figure 4:** Diurnal profiles of ClNO₂ and its photolysis rate constant, $J_{ClNO2}$, for (a) the Red Sea and (b) the Gulf of Oman. The solid lines represent the median values, the shaded areas correspond to the 10th and 90th percentiles.



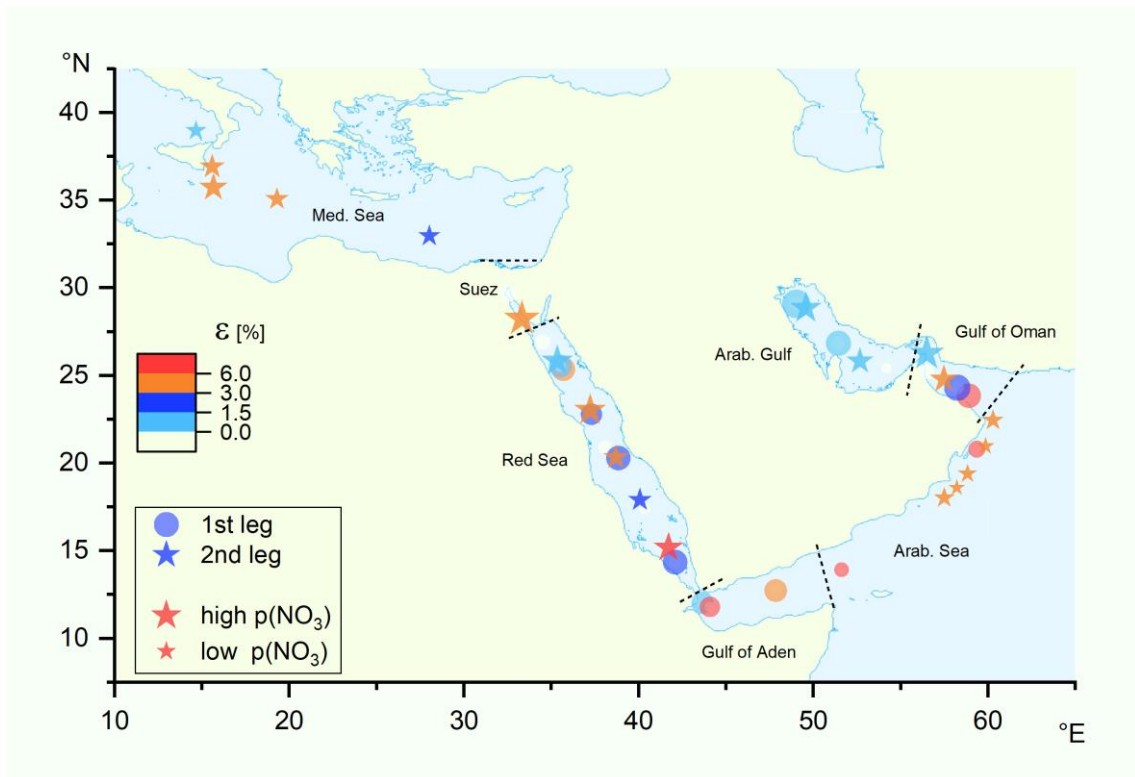

**Figure 5:** Median night-time values of ε (the ClNO₂ yield per NO₃ molecule formed) calculated via Eq. (4). The size of the symbols represents (logarithmic scale) the median NO₃ production rate, $p(NO_3)$, ranging from 0.001 pptv s$^{-1}$ in the Arabian Sea to 1.2 pptv s$^{-1}$ in the Suez Canal. Different regions are separated by the dashed black lines.



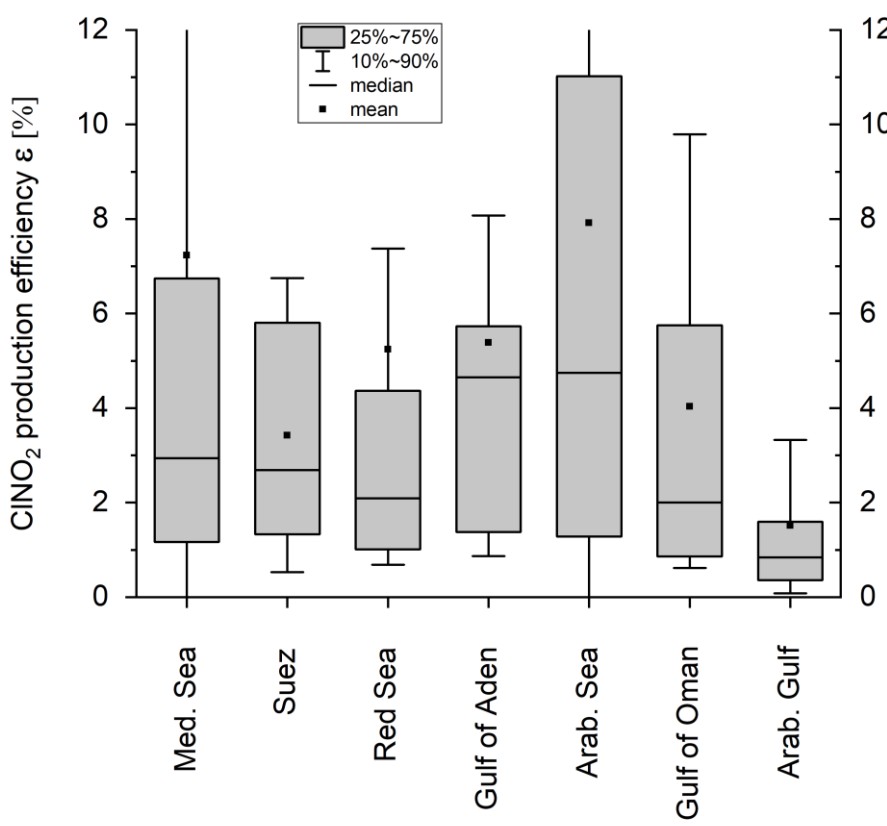

**Figure 6:** Box plots of ε (ClNO₂ production efficiency) for different regions, based on all corresponding individual night-time values calculated from Eq. (4).



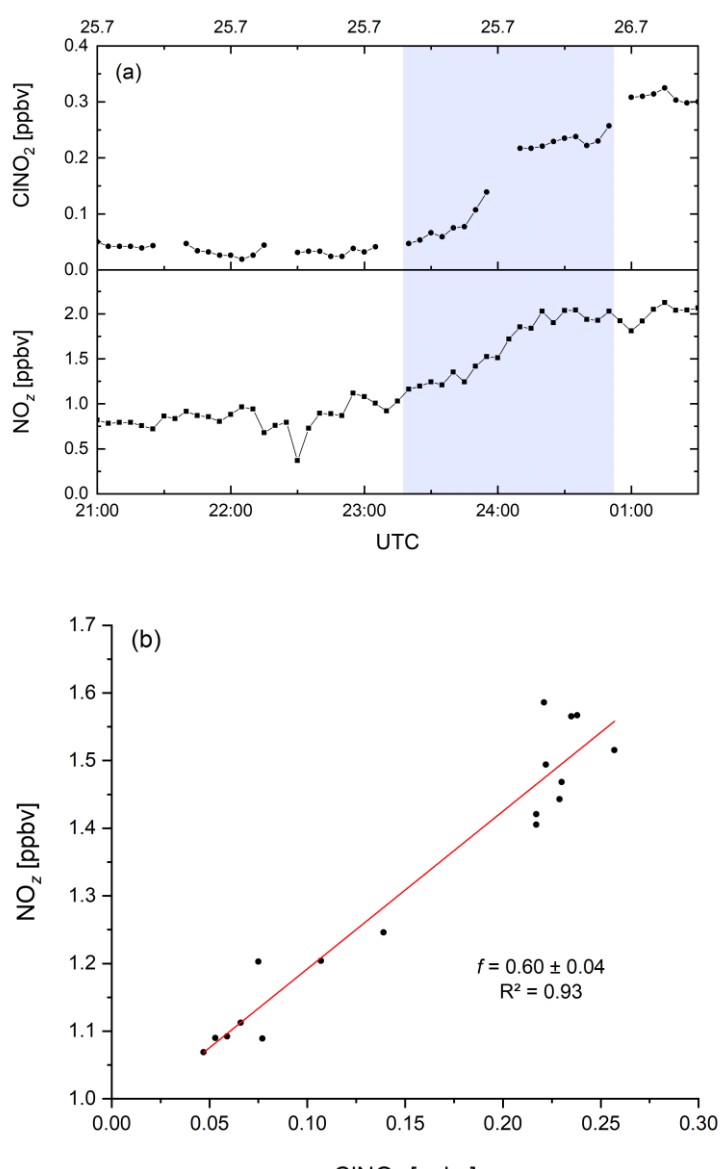

**Figure 7:** (a) Time series of $ClNO_2$ and $NO_z$ on the 25.–26.07 in the Gulf of Oman. (b) The slope of the plot of $NO_2$ vs. $ClNO_2$ can be used (Eq. 9) to calculate $f = 0.60 \pm 0.04$ for the ~ 2 h period (blue shaded area in Fig.7a).



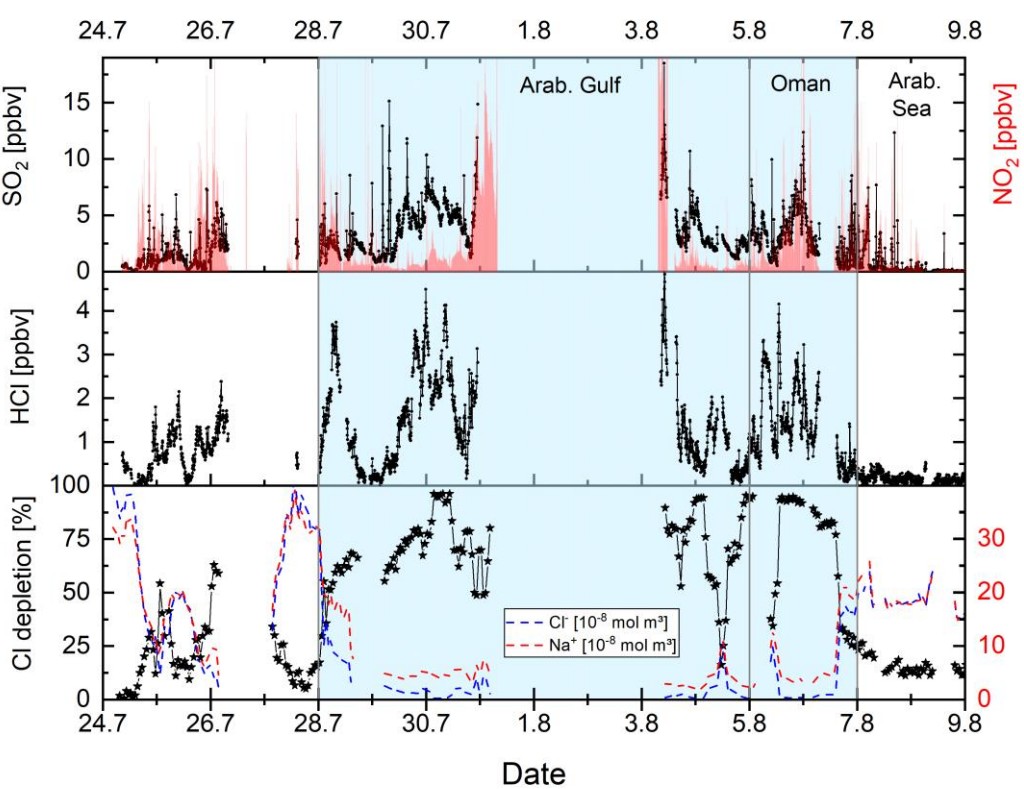

**Figure 8:** Co-variance between mixing ratios of $SO_2$, $NO_2$ and HCl and particulate chloride depletion (calculated from Eq. 12) illustrated by the difference in $Cl^-$ and $Na^+$ measured. Chloride depletion of up to 90 % indicates effective acid displacement of HCl by $HNO_3$ and $H_2SO_4$ in this region.





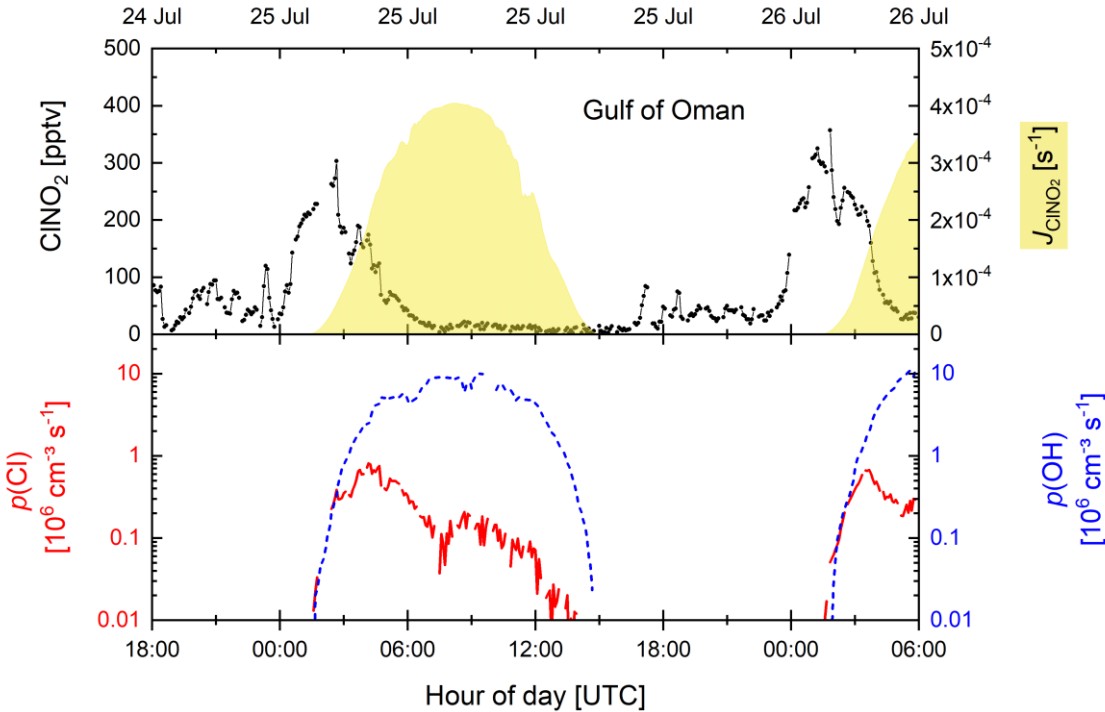

**Figure 9:** Time series of ClNO₂ mixing ratios, $J_{\text{ClNO2}}$ photolysis rates and production of Cl-radicals from ClNO₂ photolysis and OH-radicals from O₃ photolysis in the presence of H₂O ffor two consecutive nights in the Gulf of Oman.





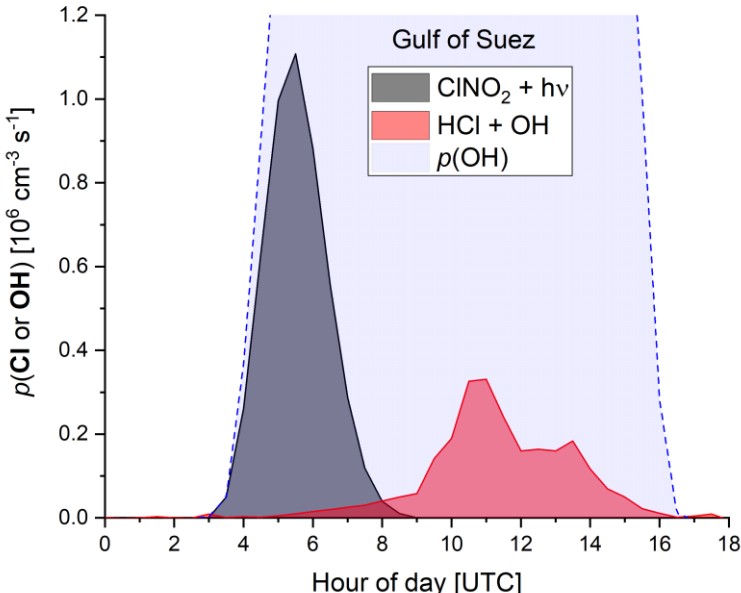

**Figure 10:** Production of Cl-atoms (from ClNO$_2$ photolysis and HCl + OH) and OH-radicals (from O$_3$ photolysis) over one diurnal cycle in the Suez Canal / Gulf of Suez.