# Peer review of "Shipborne measurements of ClNO2 in the Mediterranean Sea and around the Arabian Peninsula during summer"

_Atmospheric Chemistry and Physics, 2019_

## Referee Comment (RC1) · Anonymous Referee #1 · 4 Aug 2019

Eger et al. present measurements of ClNO2, HCl, SO2, O3, and NO2 mixing ratios in the Eastern Mediterranean Sea around the Arabian Peninsula during summer 2017. The data inform about the conversion of N2O5 to ClNO2 in this environment and are a welcome addition to the literature as there are few such measurements outside North America, China and Europe. The analysis is thorough and shows that the ClNO2 production from NO3 (epsilon) is small, which is not surprising considering the warm temperatures that shift the equilibrium away from N2O5 chemistry.

The data set is broken up into sections by region (Gulf of Oman, Gulf of Aden, Red Sea, etc.) and presented as (nocturnal) averages, median, and maxima (Table 1 + Figure 2).

[Figure]

I felt that this wasn't the most accessible way to present the data and added confusion. Examples are the averaged nocturnal locations shown in Figs 2 and 5. It would have been more transparent to present a continuous trace of ship locations color-coded by ClNO2 mixing ratios (and split the figure up into part a = leg 1 and part b = leg 2). After all, averages can be skewed by plumes. I also felt that the data were over-interpreted since changes are interpreted as regional differences rather than temporal ones. Imo, it would have perhaps been more informative to stay put in one or two places for some time for that reason, but that was perhaps outside the control of authors.

Overall, the manuscript is suitable for ACP. However, the manuscript is on the long side and could (and perhaps should) be condensed Some figures are missing axis labels (Latitude, longitude, day, month, etc.). There are also few typos in equations that need to be corrected. The date formatting does not adhere to ACP standards, e.g., 25 July 2007 (dd month yyyy), not 25.7.

Specific comments:

Page 2, reaction (6). There is also a minor channel producing NO2.

Page 4 line 29. You would get two time series, one for m/z 208 and another for m/z 210. Please specify how you used two masses (added them, or averaged them) or did you chose one over the other? Note that you can get a IBr- at 208.

Line 31 – here, you give one sensitivity. Shouldn't the sensitivity at m/z 210 be ∼1/3 that of m/z 208?

Page 5- line 2. Again, how did you use two m/z values to get one mixing ratio?

Page 5. Please comment if the stack emissions truly interfered with ClNO2 measurement by CIMS, or if the data were filtered simply as a precaution.

Page 6 line 12 "modified" how? Was it equipped with a photolytic converter?

Page 6 lines 18-19. Meusel et al. 2016 state that J values were not corrected for

up-welling UV radiation. This should also be stated here since it biases the J values low.

Page 9 equation (2). [O3] also changes over time.

Page 10 "Boundary layer height of 1000 m". That seems high for the marine boundary layer. Is there evidence to corroborate such a high mixing height?

Page 11 equation (8). Please define Keq. Should it be capitalized?

Page 13 Please check equation 11 – does not look right.

Line 2 – "k" does not appear in equation 11; perhaps it should say "B" rather than "Bk"?

Line 10. 14+/-14% and throughout the paper. Since the % operator applies to both 14 and 14, please add brackets (14+/14)%. Otherwise, it reads as a percent error, i.e., is interpreted as 14+/-2. This is repeated throughout the paper (e.g., line 21, 13+/10% could be 13+/-1.3). Same goes for units when uncertainties are given (e.g., line 31, 25-35 °C)

Page 15 line 15 – punctuation error. Note that there are others like this throughout the paper.

Equation 15 is incorrect.

Page 19. Many references are missing doi's.

Figure 2 and Figure 3. Please label all axes for clarity.

Figure 4 A lot of the variability may be due to not having enough data. Consider longer averages (1 hr, 90 min, or 2 hr) for the ClNO2 data.

Figure 7. Are you sure if this analysis is valid? It is possible that changes in NOz and ClNO2 are due to shifting air mass.

Figure 8. Are these total chloride and total sodium concentrations, or from certain size fractions only? (state in caption)

Figures 9 and 10. I think it's important to point out here that the authors only consider selected sources of radicals (ClNO2 photolysis and O1D+H2O). Some important ones are omitted (such as Cl2 and HONO photolysis and HO2+NO).

Figure 10 I wouldn't lump HCl+OH->H2O+Cl (a conversion of one radical to another) in with OH and Cl production from O3 and ClNO2 photolysis (which generate radicals from stable molecules).

[Figure]

---

## Referee Comment (RC2) · Anonymous Referee #2 · 6 Aug 2019

This paper (Shipborne measurements of ClNO2 in the Mediterranean Sea and around the Arabian Peninsula during summer) reports observations of ClNO2, NO3/N2O5, HCl, particle composition and other parameters made during a cruise in the Mediterranean Sea, Red Sea and Persian Gulf. This is a severely understudied region in terms of atmospheric chemistry and, as such, the dataset presented here fills a significant gap. The paper is well laid out, the figures and tables clear, and the analysis of the data is interesting and thorough. I only have a few minor observations, but other than that, I recommend publications on ACP.

Main Points

[Figure]

I find the analysis in Section 3.4 a bit confused. First of all, a little introduction explaining how the factors influencing ClNO2 production efficiency are going to be evaluated in this section would be useful in order to follow the discussion. Second, the values of f calculated with Eq 9 and with Eq 10 are significantly different, but this discrepancy is not really explained or discussed. It is also not clear if the value for the Gulf of Oman is 0.6 (page 11, line 29) or 0.84 (page 12, line 3).

When it comes to f, the main issue is the availability of particulate chloride. In general, it seems (page 12, lines 20-25) that the authors are focusing on fine particles, while I would expect sea salt to be a dominant source of chloride in the open sea. It may be true that the surface area of sea salt is smaller but the ClNO2 yield is higher, as the authors themselves acknowledge on page 13. Therefore neglecting sea salt in the calculation of f may not be appropriate and could possibly lead to a bias in the results of the analysis.

Finally the statement on page 13 line 30 about the importance of kdir, i.e. the direct NO3 loss, seems to be in contrast with the last lines of the section. I am afraid it is not enough to refer to a future publication, given that a significant part of the analysis stands on the assumption that the direct losses of NO3 dominate over the indirect losses. At least a summary of the steady state analysis mentioned here should be given to support the statements about kdir.

Minor Points

page 1, line 30: capitalize "Earth"

page 5, line 2: I am not sure I follow the ion chemistry from HCl to I(CN)Cl-. Where is the CN group coming from? Please provide more information or add the relevant reference.

page 5, line 7 and 12: can you provide more information on the purpose of the IMR bypass? And it is not clear to me how 50 cm of a 1/8 inch tube reduces the pressure

in a 3 m long inlet.

page 6, line 10: do you mean NO3?

page 10: can you specify which of the methods explained in the supplement is being used as default in the paper discussion and in Figure 6? I am guessing method B but it should be stated.

equation 8: I think you need to explain the keq[NO2] part of the equation and how it is related to [N2O5].

At several points in the paper the notation ICINO2- (or similar) is used for the masses measured by CIMS. But ICINO2 is a cluster not a molecule, so it should be more correctly indicated as I.CINO2-. The same for other ions mentioned throughout the paper.

---

## Author Comment (AC1) · 15 Aug 2019

**Reply to RC1**

*In the following, the referee's comments are reproduced (black) along with our replies (blue) and changes made to the text (red) in the revised manuscript.*

**General statement:**

Eger et al. present measurements of ClNO2, HCl, SO2, O3, and NO2 mixing ratios in the Eastern Mediterranean Sea around the Arabian Peninsula during summer 2017. The data inform about the conversion of N2O5 to ClNO2 in this environment and are a welcome addition to the literature as there are few such measurements outside North America, China and Europe. The analysis is thorough and shows that the ClNO2 production from NO3 (epsilon) is small, which is not surprising considering the warm temperatures that shift the equilibrium away from N2O5 chemistry.

We thank the referee for the positive evaluation of our manuscript and the useful comments and suggestions. We modified the manuscript according to the comments listed below.

**General comments:**

The data set is broken up into sections by region (Gulf of Oman, Gulf of Aden, Red Sea, etc.) and presented as (nocturnal) averages, median, and maxima (Table 1 + Figure 2). I felt that this wasn't the most accessible way to present the data and added confusion. Examples are the averaged nocturnal locations shown in Figs 2 and 5. It would have been more transparent to present a continuous trace of ship locations color-coded by ClNO2 mixing ratios (and split the figure up into part a = leg 1 and part b = leg 2).

In line with these suggestions we have redrawn Fig. 2 which now shows a continuous trace of nocturnal data points colour-coded by $ClNO_2$ mixing ratios and split up into first and second leg. To avoid excessive overlap of data points we use 1-hour averages instead of the original 5 min data. The former Fig. 2 was shifted to the supplement (now Fig. S6) and the manuscript text has been modified:

"Maximum $ClNO_2$ mixing ratios observed during each night ranged from the limit of detection to 586 pptv (see Fig. S6 for details). Figure 2 shows 1-hour averaged $ClNO_2$ mixing ratios along the ship track during (a) first and (b) second leg. Text boxes indicate the median night-time mixing ratios of $O_3$, HCl, $NO_2$ and $SO_2$ for the different regions where data from the first and second leg datasets have been combined."

We also added a similar plot color-coded by ε (complemental to Fig. 5) to the supplement to provide additional information. A reference was added to the text.

"[…] (for a more detailed plot with 1-hour averaged data points see Fig. S13)."

After all, averages can be skewed by plumes.

The nocturnal averages of ε in Fig. 5 are median values and thus less influenced by single plumes than mean values would be.

I also felt that the data were over-interpreted since changes are interpreted as regional differences rather than temporal ones. Imo, it would have perhaps been more informative to stay put in one or two places for some time for that reason, but that was perhaps outside the control of authors.

10 The aim of the AQABA campaign was to achieve a large spatial coverage around the Arabian Peninsula (within a reasonable time period of 2 months) as the whole region is severely understudied. It was thus not intended and out of the control of the authors to stay longer in some area to increase statistics. Despite high temporal variability in observed $ClNO_2$ mixing ratios within one region, we still think that the separation into different regions is a useful way

15 to present the large amount of data and is not unjustified with respect to the different air mass characteristics (e.g. $NO_X$ levels).

Overall, the manuscript is suitable for ACP. However, the manuscript is on the long side and could (and perhaps should) be condensed.

20 The analysis of this large dataset covering a two-month campaign is quite complex and the paper is necessarily a bit on the long side. To keep it as short as possible we already had moved information (e.g. details of the calculation of ε and corrections to the aerosol particle surface area concentration) to the supplement and felt we had the right balance of information in the main manuscript and the supporting information. We have not been able to identify

25 further text sections we could easily move to the supplement without perturbing the basic structure of the manuscript. However, we shifted Fig. 7 (calculation of *f* via first method, now Fig. S14) to reduce the amount of figures.

Some figures are missing axis labels (Latitude, longitude, day, month, etc.).

30 We added labels to the figures where they were missing, see specific comments below.

There are also few typos in equations that need to be corrected.

Typos in equations have been corrected, see specific comments below.

35 The date formatting does not adhere to ACP standards, e.g., 25 July 2007 (dd month yyyy), not 25.7.

We changed the date formatting throughout the manuscript, see specific comments below.

**Specific comments:**

Page 2, reaction (6). There is also a minor channel producing NO2.

We added the reaction for the minor production channel and changed the labelling of the two linked reactions. We now write:

$NO_3 + hv$ $\rightarrow$ $NO + O_2$ (R6a)

$NO_3 + hv$ $\rightarrow$ $NO_2 + O$ (R6b)

Page 4 line 29. You would get two time series, one for m/z 208 and another for m/z 210. Please specify how you used two masses (added them, or averaged them) or did you chose one over the other?

We chose $m/z$ 208 for its higher S/N-ratio to calculate the $ClNO_2$ mixing ratios. The mean ratio of $m/z$ 208 to $m/z$ 210 for the whole campaign was 3.08 (which is very close to the theoretical value of 3.13) with a correlation coefficient of $R^2$ = 0.96. We added the corresponding plot to the supplement and modified the text with the following:

"We chose the signal at $m/z$ 208 for its higher signal-to-noise (S/N) ratio to calculate the $ClNO_2$ mixing ratios reported. For the whole campaign dataset, the ratio between $m/z$ 208 and $m/z$ 210 was 3.08 ($R^2$ = 0.96, see Fig. S1) which is very close to the expected value of 3.13 derived from the natural abundance of the $^{35}Cl$ and $^{37}Cl$ isotopes, indicating no significant interferences at either of the two $m/z$."

Note that you can get a IBr- at 208.

According to Liao et al. (2011), $HO^{81}Br$ can also be detected as $I^{81}Br^-$, possibly interfering with $IClNO_2^-$ at $m/z$ 208. However, this is a minor channel compared with the formation of $IHO^{81}Br^-$ ($m/z$ 223) and, as stated above, the correlation between $m/z$ 208 and $m/z$ 210 was very good, indicating that we did not detect any significant interference.

Liao, J., et al. "A comparison of Arctic BrO measurements by chemical ionization mass spectrometry and long path-differential optical absorption spectroscopy." *Journal of Geophysical Research: Atmospheres* 116.D14 (2011).

Line 31 – here, you give one sensitivity. Shouldn't the sensitivity at m/z 210 be ~1/3 that of m/z 208?

Yes, that is true. We added the sensitivity for $m/z$ 210 to the text, which is 0.20 Hz pptv$^{-1}$.

"$I \cdot ClNO_2^-$ is more specific than $ICl^-$ ($m/z$ 162 and 164) and has a lower background signal, providing a sensitivity of 0.61 Hz pptv$^{-1}$ per $10^6$ Hz of $I^-$ at $m/z$ 208 (and 0.20 Hz pptv$^{-1}$ at $m/z$ 210), a limit of detection (LOD) ($2\sigma$, 5 min) of 12 pptv and a total measurement uncertainty of 30 % ± 6 pptv."

Page 5- line 2. Again, how did you use two m/z values to get one mixing ratio?

We only used *m/z* 188 to calculate the HCl mixing ratio, as *m/z* 190 suffers from a high background signal and an interference from I·HNO$_3^-$. We added the sensitivity for *m/z* 190 to the text, which is 0.05 Hz pptv$^{-1}$.

"HCl was observed as I(CN)Cl$^-$ (*m/z* 188 and 190) (Eger et al., 2019) with a sensitivity of 0.17 Hz pptv$^{-1}$ per 10$^6$ Hz of I$^-$ at *m/z* 188 (and 0.05 Hz pptv$^{-1}$ at *m/z* 190), a detection limit of 98 pptv and a total measurement uncertainty of 20 % ± 72 pptv. As *m/z* 190 suffers from known interferences (e.g. I·HNO$_3^-$) and has a lower S/N ratio, we used *m/z* 188 to calculate the HCl mixing ratios reported."

Page 5. Please comment if the stack emissions truly interfered with ClNO2 measurement by CIMS, or if the data were filtered simply as a precaution.

The datasets of all instruments sampling from the common inlet were filtered for our own stack emissions as a precaution because these fresh emissions (containing large amounts of particles, NO$_x$, hydrocarbons, black carbon, soot etc.) superimpose with the measured air masses and can potentially bias the results. Data where NO is above background level (like in our own ship's plume) is excluded from the calculation of ε, anyway. We amended the text:

"All datasets were filtered prior to analysis for periods where the measurements were contaminated by stack emissions to avoid a potential bias in the results."

Page 6 line 12 "modified" how? Was it equipped with a photolytic converter?

We agree that the word "modified" adds confusion, so we decided to remove it and to add a reference instead (Li et al., 2015), describing the instrument with its modifications.

"NO and NO$_2$ were measured by a chemiluminescence detector (CLD 790 SR, ECO Physics, Duernten, Switzerland) (Fontijn et al., 1970; Li et al., 2015)."

Page 6 lines 18-19. Meusel et al. 2016 state that J values were not corrected for up-welling UV radiation. This should also be stated here since it biases the J values low.

That is right, the J-values were not corrected for upwelling UV radiation, which is included in the overall uncertainty. We added a note to the text:

"J-values were not corrected for upwelling UV radiation and are estimated to have an overall uncertainty of ≈ 10 %."

Page 9 equation (2). [O3] also changes over time.

In our simple calculation of ε we assume that [O$_3$] does not change over time, as already mentioned in the text. The relative decrease in [O$_3$] over time is usually negligible (< 10 %), given the total uncertainty of the calculation, whereas the relative increase in [NO$_2$] from time $t_0$ to $t$ can be large and has to be accounted for. We added a line to emphasise this:

"In this calculation we assume that [NO$_2$] changes over time but [O$_3$] stays constant in good approximation."

Page 10 "Boundary layer height of 1000 m". That seems high for the marine boundary layer. Is there evidence to corroborate such a high mixing height?

We used an estimated boundary layer height only to give an example of potential rates of $HNO_3$ loss, which would impact on our calculation of the reaction time via equation (5). We also state that we relax the criterion for a match between calculated time and time elapsed since the beginning of the night as such effects, which depend i.a. on the boundary layer height and the $HNO_3$ to $NO_2$ ratio, are rather uncertain. We have deleted the reference to boundary layer height as it was NOT used in correction, and may have been misleading.

Page 11 equation (8). Please define Keq. Should it be capitalized?

Yes, $K_{eq}$ should be capitalized. We added a definition to the text.

"[…] where $A$ is the particle surface area concentration, $\bar{c}$ is the mean molecular velocity of $N_2O_5$ ((24400 ± 160) cm s$^{-1}$ during AQABA) and $K_{eq} = \frac{[N_2O_5]}{[NO_2]\,[NO_3]} = 2.8 \times$ 10$^{-27}$ (T/300)$^{-0.6}$ exp(11000/T) cm$^3$ molecule$^{-1}$ (IUPAC, 2019) is the temperature-dependent equilibrium constant (Reactions R4 and R5)."

Page 13 Please check equation 11 – does not look right.

Line 2 – "k" does not appear in equation 11; perhaps it should say "B" rather than "Bk"?

The equation was corrected (brackets were placed the wrong way) and parameters were renamed for better readability.

$$\gamma = B \times k \times \left( 1 - \left( \left( a \times \frac{[H_2O(l)]}{[NO_3^-]} \right) + 1 + \left( b \times \frac{[Cl^-]}{[NO_3^-]} \right) \right)^{-1} \right) \tag{11}$$

where $B = 3.2 \times 10^{-8}$ s, $k = 1.15 \times 10^6 - 1.15 \times 10^6$ exp(-0.13 [$H_2O(l)$]) s$^{-1}$ is the rate constant for the reaction $N_2O_5(aq) + H_2O(l)$, $a = 0.06$ denotes the ratio of rate constants for reactions $H_2NO_3^+(aq) + H_2O(l)$ and $H_2NO_3^+(aq) + NO_3^-(aq)$ and $b = 29$ denotes the ratio of rate constants for reactions $H_2NO_3^+(aq) + Cl^-$ and $H_2NO_3^+(aq) + NO_3^-(aq)$.

Line 10. 14+/-14% and throughout the paper. Since the % operator applies to both 14 and 14, please add brackets (14+/14)%. Otherwise, it reads as a percent error, i.e., is interpreted as 14+/-2. This is repeated throughout the paper (e.g., line 21, 13+/10% could be 13+/-1.3). Same goes for units when uncertainties are given (e.g., line 31, 25-35 °C)

This was corrected throughout the manuscript.

Page 15 line 15 – punctuation error. Note that there are others like this throughout the paper.

Punctuation errors were corrected throughout the manuscript.

Equation 15 is incorrect.
We corrected the typo in the equation:

$$pOH_{O3} = \frac{2\,J_{O1D}\,[O_3]\ \times\ k_{H2O}\,[H_2O]}{k_{H2O}\,[H_2O] + k_{N2}\,[N_2] + k_{O2}\,[O_2]}$$

Page 19. Many references are missing doi's.
DOIs have been added.

Figure 2 and Figure 3. Please label all axes for clarity.
Missing labels have been added to the figures. Dates have been modified (e.g. "Day in July").

Figure 4 A lot of the variability may be due to not having enough data. Consider longer averages (1 hr, 90 min, or 2 hr) for the ClNO2 data.
The variability is mainly due to a mixture of high atmospheric variability and a limited number of days we spent in one region. Changing the averaging interval to e.g. 1 hour did not significantly change the shape of the curves.

Figure 7. Are you sure if this analysis is valid? It is possible that changes in NOz and ClNO2 are due to shifting air mass.
(Note: The former Fig. 7 was shifted to the supplement and is now Fig. S14, see above.)
For this analysis we assume (as stated in the text) that we sample a homogeneous air mass (indicated by wind direction, T, RH etc.), i.e. changes in $NO_z$ and $ClNO_2$ are not caused by a change of the air mass within the period of observation. As this requirement was rarely fulfilled, we could only analyse the four different episodes mentioned in the text.

Figure 8. Are these total chloride and total sodium concentrations, or from certain size fractions only? (state in caption)
(Note: The former Fig. 8 is now Fig. 7.)
These are $PM_1$ data only (AMS measurements, see Sect. 2.5). We added the information to the caption:
"Co-variance between mixing ratios of $SO_2$, $NO_2$ and HCl and particulate chloride depletion (calculated from Eq. 12) illustrated by the difference in $Cl^-$ and $Na^+$ ($PM_1$) measured."

Figures 9 and 10. I think it's important to point out here that the authors only consider selected sources of radicals (ClNO2 photolysis and O1D+H2O). Some important ones are omitted (such as Cl2 and HONO photolysis and HO2+NO).
We added text to the beginning of section 3.6:
"Other potential Cl sources (e.g. $Cl_2$ photolysis) are not considered here as we do not have experimental data to quantify their impact."

We also added a sentence to the paragraph where we describe the calculation of $P_{OH}(O_3)$:
"As we do not consider other OH production channels (e.g. photolysis of HONO or $HO_2$ + NO), which can be of importance under more polluted conditions, $pOH_{O3}$ represents a lower limit of $pOH$."

The labelling and caption of the former Fig. 9 (now Fig. 8) has been modified to emphasise that only exclusive channels were considered:
"Time series of $ClNO_2$ mixing ratios, $J_{ClNO2}$ photolysis rates and production of Cl-radicals from $ClNO_2$ photolysis ($pCl_{ClNO2}$) and OH-radicals from $O_3$ photolysis in the presence of $H_2O$ ($pOH_{O3}$) for two consecutive nights in the Gulf of Oman."

Figure 10 I wouldn't lump HCl+OH->H2O+Cl (a conversion of one radical to another) in with OH and Cl production from O3 and ClNO2 photolysis (which generate radicals from stable molecules).

In former Fig. 10 (now Fig. 9) we only compare the relative contribution of HCl + OH and $ClNO_2$ + hv to Cl radical formation. Although transformation of OH into Cl does not change the overall radical budget nor the $RO_2$ budget, the relative oxidation rates of several VOCs will be modified as Cl reacts much faster with some of them than OH does.

---

## Author Comment (AC2) · 15 Aug 2019

**Reply to RC2**

*In the following, the referee's comments are reproduced (black) along with our replies (blue)*
*and changes made to the text (red) in the revised manuscript.*

**General statement:**

This paper (Shipborne measurements of ClNO2 in the Mediterranean Sea and around the
Arabian Peninsula during summer) reports observations of ClNO2, NO3/N2O5, HCl, particle
composition and other parameters made during a cruise in the Mediterranean Sea, Red Sea
and Persian Gulf. This is a severely understudied region in terms of atmospheric chemistry
and, as such, the dataset presented here fills a significant gap. The paper is well laid out, the
figures and tables clear, and the analysis of the data is interesting and thorough. I only have a
few minor observations, but other than that, I recommend publications on ACP.
We thank the referee for the positive evaluation of our manuscript and the useful comments
and suggestions. We modified the manuscript according to the comments listed below.

**General comments:**

I find the analysis in Section 3.4 a bit confused. First of all, a little introduction explaining how
the factors influencing ClNO2 production efficiency are going to be evaluated in this section
would be useful in order to follow the discussion.
We added an introductive sentence to the section:
"In the following we calculate $f$ and $\gamma$ from our measurements, compare the values with the
literature and quantify the contributions of $k_{het}$ and $k_{dir}$ to the overall NO$_3$ loss rate."

Second, the values of f calculated with Eq 9 and with Eq 10 are significantly different, but this
discrepancy is not really explained or discussed.
With Eq. 10 we calculated median values of $f$ for each region, based on all available data, and
listed them in Table 2 (here we added a note: "[a] Calculated from Eq. (10)"). In contrast, Eq. 9
could only be applied to four specific time periods listed in the text, when we sampled a
homogeneous air mass. There is no reason to expect a perfect agreement as these four
values calculated with Eq. 9 are only snapshots within different regions.

It is also not clear if the value for the Gulf of Oman is 0.6 (page 11, line 29) or 0.84 (page 12,
line 3).
These are two values for two different time periods, both in the Gulf of Oman (first one shown
in Fig. S14 (formerly Fig. 7), second one not shown). We added a date to the first event to
make this clear.

"[…] as illustrated in Fig. S14 for data obtained in the Gulf of Oman (25–26 July 2017) for which $f = 0.60 \pm 0.04$."

When it comes to f, the main issue is the availability of particulate chloride. In general, it seems (page 12, lines 20-25) that the authors are focusing on fine particles, while I would expect sea salt to be a dominant source of chloride in the open sea. It may be true that the surface area of sea salt is smaller but the ClNO2 yield is higher, as the authors themselves acknowledge on page 13. Therefore neglecting sea salt in the calculation of f may not be appropriate and could possibly lead to a bias in the results of the analysis.

We agree that sea salt may contribute to $ClNO_2$ formation (due to an $f$ close to 1). However, the average contribution from the coarse mode during the campaign was only 14 % (as stated in the text) and $f$ is between 0.5 and 1 for fine mode particles as well, so the fractional contribution of coarse mode sea salt to $ClNO_2$ formation is generally low (as most of the coarse mode particles were dust). This is discussed in detail towards the end of Sect. 3.4.

We note that, the calculation of $\varepsilon = \left( \frac{[ClNO_2]}{[NO_3]_{int}} \right)$ is in any case independent of $k_{het}$ (and thus independent of the contributions of the coarse/fine mode).

Finally the statement on page 13 line 30 about the importance of kdir, i.e. the direct NO3 loss, seems to be in contrast with the last lines of the section. I am afraid it is not enough to refer to a future publication, given that a significant part of the analysis stands on the assumption that the direct losses of NO3 dominate over the indirect losses. At least a summary of the steady state analysis mentioned here should be given to support the statements about kdir.

The last lines of the section state that much of the reactivity could neither be attributed to $k_{het}$ nor to measured VOCs but to unidentified compounds (also contributing to $k_{dir}$), so they are not in contrast with the statement that $k_{dir}$ is more important than $k_{het}$.

The result that $k_{dir \gg} k_{het}$ is also derived from measurement data via $\varepsilon = f \left( \frac{k_{het}}{k_{het} + k_{dir}} \right)$.

To make this clear, we modified the text:

"However, for a large fraction of each night NO3 was below the detection limit (ca. 5 pptv) despite a high production rate (large mixing ratios of $NO_2$ and $O_3$). A steady-state analysis of $NO_3$ production and loss indicated a high total reactivity which could not be attributed to measured trace gases ($k_{dir}$) or heterogeneous losses of $N_2O_5$ ($k_{het}$). A detailed analysis of the $NO_3$ lifetime and the role of VOCs is beyond the scope of the present manuscript and will be described in detail in a separate publication."

**Specific comments:**

page 1, line 30: capitalize "Earth"
Done.
"As the Arabian Gulf already suffers from some of the most polluted air on Earth […]"

page 5, line 2: I am not sure I follow the ion chemistry from HCl to I(CN)Cl-. Where is the CN group coming from? Please provide more information or add the relevant reference.
Detection of HCl involves $I(CN)_2^-$ primary ions. A reference (Eger et al., 2019) was added.
"HCl was observed as $I(CN)Cl^-$ (*m/z* 188 and 190) (Eger et al., 2019) with a sensitivity of 0.17 Hz pptv$^{-1}$ per $10^6$ Hz of $I^-$ at *m/z* 188 (and 0.05 Hz pptv$^{-1}$ at *m/z* 190), a detection limit of 98 pptv and a total measurement uncertainty of 20 % ± 72 pptv."

page 5, line 7 and 12: can you provide more information on the purpose of the IMR bypass?
The bypass in front of the IMR (1 slm) was used in order to improve the transmission of $CH_3C(O)O_2$ radicals from the thermal decomposition of PAN in the heated inlet (Eger et al. (2019). As this is not relevant for $ClNO_2$, HCl or $SO_2$ detection, we removed the sentence.

And it is not clear to me how 50 cm of a 1/8 inch tube reduces the pressure in a 3 m long inlet.
The pressure is both reduced by a bypass flow (5 slm) and by the mentioned (coiled) piece of 1/8 inch tube (which is reducing the pressure in the inlet line via energy dissipation to the walls). We added the missing information to the text (and changed units from inch to mm):
"To avoid condensation of water in the inlet lines in the containers, the pressure in the sampling line was reduced to ≈ 700–800 mbar with a bypass flow of ≈ 5 slm and by including an additional ≈ 50 cm long (coiled) piece of 3.18 mm (OD) PFA tubing."

page 6, line 10: do you mean NO3?
$NO_2$ is correct.

page 10: can you specify which of the methods explained in the supplement is being used as default in the paper discussion and in Figure 6? I am guessing method B but it should be stated.
Method C is used as default as stated in the supplement. We wanted to avoid confusion by referring to a method that is only mentioned in the supplement, since the data reduction is also (briefly) outlined in the manuscript itself. Nevertheless, we added a note with a reference to the supplement to Sect. 3.2.
"The data reduction is described in more detail in the supplement (all the data shown in the manuscript corresponds to the application of method C), where the sensitivity of ε to these limitations and additional constraints is discussed."

equation 8: I think you need to explain the keq[NO2] part of the equation and how it is related to [N2O5].

We added a definition to the text:

5 "[…] where $A$ is the particle surface area concentration, $\bar{c}$ is the mean molecular velocity of $N_2O_5$ ((24,400 ± 160) cm s$^{-1}$ during AQABA) and $K_{eq} = \frac{[N2O5]}{[NO2]\,[NO3]} = 2.8 \times$ 10$^{-27}$ (T/300)$^{-0.6}$ exp(11000/T) cm$^3$ molecule$^{-1}$ (IUPAC, 2019) is the temperature-dependent equilibrium constant (Reactions R4 and R5)."

10 At several points in the paper the notation IClNO2- (or similar) is used for the masses measured by CIMS. But IClNO2 is a cluster not a molecule, so it should be more correctly indicated as I.ClNO2-. The same for other ions mentioned throughout the paper.

This was corrected throughout the manuscript. We now write $I \cdot ClNO_2^-$ and $I \cdot HNO_3^-$.

---

## Author Response (AR2)

**Reply to Co-Editor:**

*In the following, the Co-Editor's comments are reproduced (black) along with our replies (blue) and changes made to the text (red) in the revised manuscript:*

Please add few more figures which may or may not show variability in the correlation between m/z 208 and 210 to provide more information about possible interference (e.g. correlation plots of different regions or days would be helpful).

In addition to the correlation plot between $m/z$ 208 and 210 for the whole AQABA campaign (Fig. S1), we inserted further correlation plots from different regions (Gulf of Oman, Gulf of Aden, Red Sea, Suez Canal and Mediterranean Sea, Fig. S1b–S1f) to indicate the absence of significant interferences at either of the two $m/z$.

The text in the manuscript has been modified:
"For the whole campaign dataset, the ratio between $m/z$ 208 and 210 was 3.08 (with $R^2$ = 0.96, see Fig. S1a of the supplementary information), which is very close to the expected value of 3.13 derived from the natural abundance of the $^{35}Cl$ and $^{37}Cl$ isotopes. In addition, correlation plots from different regions (Fig. S1b–f) indicate the absence of significant interferences at either of the two $m/z$."

The text in the supplement has been modified and new figures have been inserted:
"Figure S1: Correlation between relative $ClNO_2$ signals measured at $m/z$ 208 and 210 (with slope $m$ and intercept $y_0$) for (a) the AQABA campaign and (b–f) selected periods in the Gulf of Oman, the Gulf of Aden, the Red Sea, the Suez Canal and the Mediterranean Sea."

Revised manuscript:

[revised manuscript text omitted]

[1] Regions: Red Sea, Gulf of Oman (Oman), Arabian Gulf (Arab. Gulf) and Suez Canal / Gulf of Suez (Suez). $t$ denotes the time since sunset; $t'$ corresponds to the air mass age calculated from Eq. 5 in the manuscript NO$_{3,int}$ is the total amount of NO$_3$ produced over the course of the night and $\varepsilon$ is the ClNO$_2$ production efficiency (Eq. 4 in manuscript).